EMBO
reports

# Cysteine-reactive covalent chloro-*N*-acetamide ligands induce ferroptosis mediated cell death

Gina Gotthardt [1,7], Janik Weckesser [2,3,7], Georg Tascher[1], Sara Barros da Gama[4],
Hannah J Uckelmann [4], Shibo Sun [5], Martin P Schwalm[2,3], Thorsten Mosler [1], Giulio Ferrario [6],
José Pedro Friedmann Angeli[5], Christian Münch [6], Stefan Knapp [2,3✉] & Stefan Müller [1✉]

## Abstract

Covalent inhibitors are an attractive targeting strategy that has expanded the development of degraders to target poorly druggable proteins including the E3 ligase RNF4. We show that RNF4 is a potential vulnerability of AML. High RNF4 expression levels correlate with poor patient survival and depletion of RNF4 results in increased sensitivity of AML cells to antileukemic drugs. Therefore, we aimed to develop chemical degraders (PROTACs) of RNF4 using a known covalent RNF4 ligand (CCW16), containing a chloro-*N*-acetamide group, as well as established E3 ligands targeting CRBN or VHL. However, while CCW16 and CCW16-derived PROTACs react potently with cysteines in recombinant RNF4, in cells, CCW16 forms covalent bonds with a large number of proteins, including peroxiredoxins. Consequently, CCW16 based PROTACs do not trigger degradation of RNF4, but induce the ferroptosis marker heme oxygenase-1 and impair cell viability in a distinct, RNF4-independent, ferroptotic cell death pathway. We hypothesize that other chloro-*N*-acetamide-containing E3 ligase ligands would also induce ferroptosis. Indeed, the RNF114 ligand EN219 also strongly induces ferroptosis, suggesting that ligands harboring this electrophile induce undesired off-target toxicity.

**Keywords** AML; CCW16; Covalent PROTACs; Ferroptosis; RNF4
**Subject Categories** Autophagy & Cell Death; Post-translational Modifications & Proteolysis

## Introduction

Cells are constantly exposed to endogenous or exogenous hazards that threaten their genome or proteome integrity. Sophisticated and highly regulated DNA repair pathways safeguard genome integrity, preventing the emergence and accumulation of genetic alterations that are a hallmark of tumorigenesis (Negrini et al, 2010; Hanahan and Weinberg, 2011; Jeggo et al, 2016; Hanahan, 2022). At the same time, cancer chemotherapy largely relies on the induction of genotoxic stress in order to expose DNA damage as a vulnerability of highly proliferating and mutation-prone cancer cells (Luo et al, 2009; Dobbelstein and Sørensen, 2015; Aziz et al, 2012). However, a hallmark of cancer cells is their ability to tolerate endogenous or therapy-induced genotoxic stress, which confers resistance to DNA-damaging chemotherapies (Madhusudan and Hickson, 2005; Jurkovicova et al, 2022). An emerging strategy to break this resistance mechanism is the targeting of DNA repair and DNA damage response factors (Huang and Zhou, 2021).

Accumulating evidence suggests that post-translational modifications by ubiquitin and the small ubiquitin-related modifier (SUMO) are necessary to maintain genome stability by reversibly regulating protein function and activity (Su et al, 2020a; Yu et al, 2020). Ubiquitin is conjugated to specific lysine residues of target proteins by a multistep enzymatic cascade of E1 activating enzymes, E2 conjugating enzymes and E3 protein ligases, leading to proteasomal degradation or modulation of protein activities depending on the linkage type of the ubiquitin chain (Oh et al, 2018; Kolla et al, 2022). Similar to ubiquitination, also SUMOylation occurs via a three-step cascade involving a single dimeric E1 enzyme, an E2 conjugating enzyme and E3 SUMO ligases attaching either SUMO1 or the highly related SUMO2/3 to the target. Like ubiquitin, SUMO can be conjugated as a monomer (preferably SUMO1) or as chains by SUMO-SUMO linkages via internal lysine residues (mostly SUMO2/3). In principle, SUMOylation is a non-proteolytic modification, controlling the dynamics of protein-protein interactions (Cappadocia and Lima, 2018; Flotho and Melchior, 2013; Vertegaal, 2022; Raman et al, 2013). However, SUMO chains, which are typically formed in response to genotoxic or proteotoxic stress, can connect SUMO signaling to the ubiquitin system via SUMO-targeted ubiquitin ligases (StUbLs) (Chang et al, 2021; Jansen and Vertegaal, 2021). In mammalian cells, three StUbLs, RNF4, TOPORS, and RNF111 have been identified. Their common structural characteristic is a RING-type ubiquitin ligase domain and the presence of tandemly repeated SUMO interaction

[1]Institute of Biochemistry II, Goethe University Frankfurt, Faculty of Medicine, Theodor-Stern-Kai 7, 60590 Frankfurt am Main, Germany. [2]Institute of Pharmaceutical Chemistry, Goethe University Frankfurt, Max-von-Laue-Str. 9, 60438 Frankfurt am Main, Germany. [3]Structural Genomics Consortium (SGC), Buchmann Institute for Life Sciences, Max-von-Laue-Str. 15, 60438 Frankfurt am Main, Germany. [4]Department of Pediatrics, Goethe University Frankfurt, University Hospital, Theodor-Stern-Kai 7, 60590 Frankfurt am Main, Germany. [5]Rudolf-Virchow-Zentrum, Center for Integrative and Translational Bioimaging, Julius-Maximilian University Würzburg, Josef-Schneider-Straße 2, 97080 Würzburg, Germany. [6]Institute of Molecular Systems Medicine, Goethe University Frankfurt, Faculty of Medicine, Theodor-Stern-Kai 7, 60590 Frankfurt am Main, Germany. [7]These authors contributed equally: Gina Gotthardt, Janik Weckesser. ✉E-mail: knapp@pharmchem.uni-frankfurt.de; ste.mueller@em.uni-frankfurt.de

motifs (SIMs) enabling them to specifically target and ubiquitinate polySUMOylated substrates (Lallemand-Breitenbach et al, 2008; Perry et al, 2008; Tatham et al, 2008; Erker et al, 2013).

RNF4 is a monomeric RING-type E3 ubiquitin ligase, consisting of four SIMs located in the N-terminal region. These SIMs target RNF4 to SUMOylated proteins, and a RING finger domain in the RNF4 C-terminus catalyzes the transfer of ubiquitin from an E2 enzyme to the target (Keusekotten et al, 2014). RNF4 plays a key role in the proteasome-dependent removal of misfolded or damaged nuclear proteins and regulates the resolution of cytoplasmic stress granules (Guo et al, 2014; Gärtner and Muller, 2014; Keiten-Schmitz et al, 2020b). During DNA repair, RNF4 controls chromatin residency of DNA damage response (DDR) and repair factors, and it is critical for the reversal of stalled replication forks (Prudden et al, 2007; Galanty et al, 2012; Yin et al, 2012; Wagner et al, 2019; Keiten-Schmitz et al, 2020a; Ding et al, 2022). In addition, RNF4 mediates the proteasomal clearance of SUMOylated post-replicative DNA-protein crosslinks (DPCs), including TOP1/2- (DNA topoisomerase 1/2) and DNMT1- (DNA-methyltransferase 1) DPCs, which are induced by topoisomerase poisons or hypomethylating agents (Liu et al, 2024; Sun et al, 2020; Weickert et al, 2023).

Given that RNF4 maintains proteome and genome stability and that it is overexpressed in different tumor entities such as acute myeloid leukemia (AML), RNF4 may represent a vulnerability for cancer (Chang et al, 2021; Gärtner and Muller, 2014; Ding et al, 2022). As recently shown, the depletion of RNF4 in mice expressing the oncogene *c-myc* prolongs tumor-free survival, providing a compelling rationale for targeting RNF4 as an anticancer therapy (Her et al, 2024).

With the advent of targeted protein degradation (TPD), a new pharmacological modality has become available that allows the targeting of poorly druggable proteins. One class of TPD compounds are PROteolysis TArgeting Chimeras (PROTACs), consisting of two ligands, coupled by a short linker. One ligand interacts with the protein of interest (POI) while the other one recruits an E3 ubiquitin ligase, mostly CRBN (Cereblon) or VHL (Von Hippel-Lindau) (Chirnomas et al, 2023; Diehl and Ciulli, 2022). The close proximity of the POI and the E3 ligase leads to the ubiquitination of the substrate, followed by proteasomal degradation (Békés et al, 2022). Since RNF4 is a small protein with a predicted very long unstructured N-terminal region and only a small structured domain at the C-terminus, it is challenging to develop ligands targeting selectively RNF4 (Liew et al, 2010). However, the usage of covalent binders may offer a strategy to target exposed cysteine residues to reactive ligands (Kiely-Collins et al, 2021). Recently, a cysteine-reactive compound (CCW16) has been identified by activity-based profiling (Ward et al, 2019). Linking CCW16 to JQ1, a BET (Bromo- and extra-terminal domain) inhibitor (Filippakopoulos et al, 2010), results in the development of the RNF4-targeting BRD4 degrader CCW28-3, presumably acting through RNF4 ubiquitin ligase activity (Ward et al, 2019). Thus, we anticipated that by utilizing the CCW16 warhead, degraders of RNF4 could be designed by linking this cysteine-reactive moiety to established E3 ligase ligands or to itself (homo-PROTACs) (Diehl et al, 2024).

In this study, we demonstrated that RNF4 represents a cancer vulnerability and potential drug target in AML. Depletion of RNF4 exacerbated the sensitivity of OCI-AML2 cells towards the antileukemic drug decitabine (also known as 5-AzadC, 5-Aza-2′-deoxycytidine) and the replication stress-inducing agent

aphidicolin. Decitabine, a cytosine analog, is incorporated into the DNA during replication, leading to the formation of a DNMT1-DPC upon DNA methylation, whereas aphidicolin inhibits the DNA polymerase α and δ, blocking the cell cycle at early S-phase (Santi et al, 1984; Short and Kantarjian, 2022; Krokan et al, 1981). To establish a small molecule degradation system for RNF4 depletion in AML, we used the recently published RNF4 ligand CCW16 (Ward et al, 2019). To develop heterobifunctional PROTACs recruiting the E3 ubiquitin ligases VHL or CRBN to RNF4, we used the established VHL- and CRBN-based E3 ligands and different linker moieties for proteasomal degradation of RNF4. However, a large diversity of synthesized PROTACs did not result in the degradation of RNF4 in different cell lines that were investigated by Western blotting. We studied, therefore, CCW16 target engagement in vitro as well as in cells. Comprehensive mapping of CCW16 interaction by mass spectrometry, as well as site-directed mutants, revealed covalent binding to the unstructured region of RNF4. However, in cellulo interaction studies demonstrated covalent binding to diverse cysteine-containing cellular proteins, including proteins of the peroxiredoxin family, thus identifying CCW16 as a non-selective covalent ligand. Interestingly, treatment of cells with CCW16 and CCW16-derived PROTACs revealed an upregulation of the ferroptotic marker heme oxygenase-1 (HMOX1) (Wu et al, 2022) and an increase in lipid peroxidation, affecting cell viability. Intriguingly, also the E3 ligase ligand EN219 that has been described to target the RING-type ligase RNF114 via a chloro-N-acetamide electrophile phenocopied the cellular effects of CCW16. Based on these data, we propose that treatment of cells with chloro-N-acetamide-containing compounds may lead to the activation of a target-independent, ferroptotic cell death pathway.

## Results

### RNF4 represents a vulnerability in AML cells

SUMO signaling is upregulated in many tumor entities, including hematological malignancies (Boulanger et al, 2019). Mining of TGCA expression datasets revealed an upregulation of RNF4 in leukemic cells of AML patients (Fig. 1A) (Bartha and Győrffy, 2021). Intriguingly, high expression of RNF4 correlated with poor survival of AML patients, thus defining RNF4 as a potential drug target (Fig. 1B) (Chandrashekar et al, 2017, 2022). To test this hypothesis, we investigated the dependence of AML cell lines on RNF4 using various cell-based phenotypic assays.

First, we performed a CRISPR/Cas9 dropout experiment in Cas9-expressing OCI-AML3 (Fig. 1C) and MV4-11 (Fig. 1D) cell lines and measured the cell survival over 15 days after lentiviral transduction with three different sgRNAs targeting RNF4 or a non-targeting control. We observed a drastic decrease in cell viability, especially after transduction with sgRNF4_3 in both cell lines, whereas transduction with sgRNF4_1 affected viability only in MV4-11 cells (Fig. 1C,D). In contrast, sgRNF4_2 did not have an effect on the viability of both cell lines tested (Fig. EV1A,B). Immunoblot analysis of RNF4 levels in OCI-AML3 Cas9 and MV4-11 Cas9 cells demonstrated that the reduced viability correlated with knockout (KO) efficiency. Transduction with sgRNF4_2 did not significantly reduce RNF4 levels, explaining why it did not

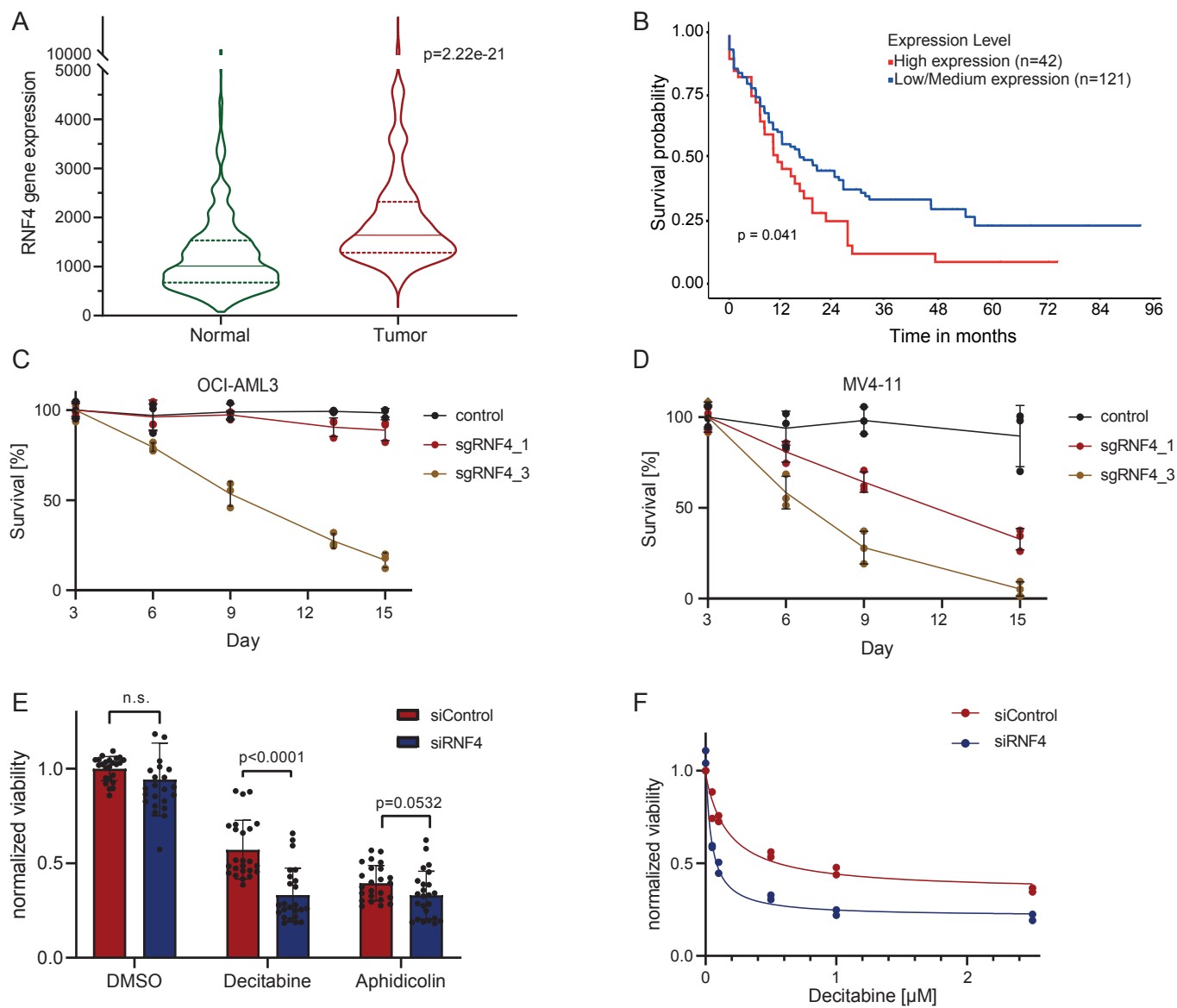

**Figure 1. RNF4 as a vulnerability of AML cells.**

(A) AML cells exhibit an elevated expression of RNF4 (derived from TNMplot.com; based on TGCA data). *P* value of Mann–Whitney *U*-test is indicated ($p = 2.22e-21$); $n = 407$ (normal), $n = 151$ (tumor). Normal: Minima = 77, Maxima = 10,063, Median = 1009, Q1 = 675, Q3 = 1533, Upper whisker = 2726; Tumor: Minima = 691, Maxima = 10,010, Median = 1640, Q1 = 1283, Q3 = 2317.5, Upper whisker = 3750. Dashed lines: 25th and 75th percentile; solid line: 50th percentile (median); Whiskers: range within 1.5×IQR from Q1/Q3. (B) Effect of RNF4 expression level on AML patient survival. High expression of RNF4 is associated with a poor prognosis (analyzed using UALCAN). *P* value of Kaplan–Meier analysis is indicated ($p = 0.041$); $n = 42$ (high expression), $n = 121$ (low expression). (C, D) RNF4 CRISPR dropout experiment in Cas9-expressing OCI-AML3 (C) or MV4-11 (D) cells. The transduction efficiency of PE-positive cells was around 50%. Survival rate was measured by flow cytometry and normalized to the empty vector control on day 3. Error bars show the standard deviation of the mean, $n = 3$ (biological replicates). (E) Cell viability assay of OCI-AML2 cells 6 days after RNF4 KD with siRNA in combination with decitabine (5-azadC) [1 µM] or aphidicolin [0.075 µg/ml]. *P* values of two-tailed, unpaired Student's *t*-tests are indicated; error bars show the standard deviation of the mean, $n = 8$ (biological replicates, each three technical replicates). DMSO: $p = 0.1754$, Decitabine: $p < 0.0001$, Aphidicolin: $p = 0.0532$. (F) Viability assay of OCI-AML2 cells after RNF4 KD with siRNA in combination with different decitabine concentrations to measure dose-response dependency. IC$_{50}$ values were determined by a nonlinear dose-response-inhibition fit, $n = 2$ (biological replicates). Source data are available online for this figure.

affect viability in the CRISPR dropout experiments (Fig. EV1C,D). By Contrast, sgRNF4_1 and sgRNF4_3 led to a reduction in RNF4 levels and the detection of a truncated form of RNF4 (Fig. EV1C,D). This truncated form was not expected to be functional since RNF4 activity relies on the C-terminal catalytic RING domain (Chiariotti et al, 1998).

## RNF4 exerts key functions in genome maintenance by mediating tolerance to replication stress or clearance of DNA-protein crosslinks

To investigate if RNF4 depletion sensitizes AML cells to genotoxic stress, we performed an siRNA-mediated knockdown (KD) of

RNF4 in OCI-AML2 cells (Fig. EV1E), followed by treatment with the DNA damage compounds decitabine and aphidicolin (Fig. 1E). We used the CellTiter-Glo® Luminescent Cell Viability Assay (CTG) to measure cell viability. In contrast to the CRISPR dropout experiment, RNF4 KD alone did not significantly affect cell viability. This observation can either be explained by the shorter time-course (3 days), transient transfection of siRNA or less efficient RNF4 depletion by siRNA-mediated KD when compared to CRISPR/Cas9-mediated KO. However, RNF4 KD in combination with the DNA damage compounds decitabine and—to a lesser extent—with aphidicolin showed synthetic lethality (Fig. 1E). The increased sensitivity of RNF4 depleted cells to aphidicolin, which induces replication stress, is consistent with the established function of RNF4 in counteracting replicative stress (Ellis et al, 2021; Oram et al, 2024; Her et al, 2024). Since decitabine treatment showed a strong decrease in cell viability after RNF4 KD compared to RNF4 KD alone, we treated OCI-AML2 with different concentrations of decitabine in combination with RNF4 KD. We detected a dose-response dependency with an $EC_{50}$ value of 0.2 μM for decitabine treatment alone and an $EC_{50}$ value of 0.04 μM for combination with RNF4 KD (Fig. 1F), demonstrating that targeting RNF4 sensitizes cells to the antileukemic drug decitabine. Decitabine induces DNMT1-DPCs, leading to DNMT1 degradation and DNA hypomethylation. Since it is well established that RNF4 is engaged in the clearance of DNMT1-DPCs, its depletion likely explains the sensitization of the cells to decitabine (Liu et al, 2021; Weickert et al, 2023).

## Design, synthesis, and validation of CCW16-based RNF4 degraders

Based on the above findings, we aimed to develop a PROTAC targeting RNF4 for degradation. PROTACs are heterobifunctional molecules that recruit a protein of interest to the ubiquitin system. The Nomura group recently identified a covalent RNF4 binder, CCW16, which was used to design PROTACs in which RNF4 acted as the recruited E3 ligase (Ward et al, 2019). We hypothesized that CCW16 could also be used as a ligand to design PROTACs leading to RNF4 degradation by recruiting E3 ligases such as CRBN and VHL, which have been established for the design of PROTACs. We therefore functionalized CCW16 as a POI recruitment ligand and designed a series of PROTACs, where CCW16 was linked to VHL032- and 4-hydroxy thalidomide-derived-E3 ligands, recruiting the E3 ligases VHL and CRBN, respectively. We synthesized four different CRBN-based (1a-d) and three different VHL-based PROTACs (2a-c) using a variety of polyethylene glycol (PEG) and alkyl linkers (Fig. 2A). The synthetic procedures and chemical characterization of all synthesized PROTACs are included in Appendix Figs. S1–S8.

To ensure cell penetration of the synthesized PROTACs, we performed NanoBRET™ (Bioluminescence Resonance Energy Transfer) assays (Figs. 2B and EV2A,B). NanoLuciferase was ectopically expressed in HEK293T cells in frame with CRBN and VHL, respectively. As expected, the synthesized compounds demonstrated target engagement with CRBN and VHL in intact or permeabilized cells with $EC_{50}$ values in the nanomolar and micromolar range, respectively. The measured $EC_{50}$ were similar to the positive control lenalidomide for CRBN-based PROTACs, demonstrating excellent cell penetration. However, $EC_{50}$ values

were in the micromolar range for VHL-based PROTACs which was significantly weaker compared with the VHL inhibitor VH298, possibly due to less efficient cell penetration of these PROTACs (Fig. EV2B).

To evaluate the potency and degradation efficiency of the candidate PROTACs, we performed Western blot analysis to detect endogenous RNF4. Initially, HeLa cells were treated for 6 h at a PROTAC concentration of 5 μM. To validate the dependence of possible degradation on the ubiquitin-proteasome system (UPS), control samples were pretreated with the proteasomal inhibitor MG-132, the ubiquitin activating enzyme (UAE) inhibitor TAK-243 or with the neddylation inhibitor MLN4924 (blocking CRBN or VHL E3 ligase activity). While inhibition of the UPS machinery stabilized RNF4, an efficient and reproducible degradation of RNF4 was not detected for any of the CRBN- or VHL-based PROTACs (Fig. 2C). To control that the anti-RNF4 antibody specifically detects RNF4 and can monitor reductions in RNF4 levels, we depleted RNF4 by directly targeting it with siRNA or by targeting SENP6, leading to auto-ubiquitylation and degradation of RNF4 (Rojas-Fernandez et al, 2014; Wagner et al, 2019) (Fig. EV2C). Further, we switched to a HeLa cell line expressing an endogenously FLAG-tagged RNF4. Cells were treated for 5, 8, and 24 h with the synthesized PROTACs at concentrations between 1 and 5 μM (Fig. EV2D), but no significant reduction of Flag-tagged RNF4 was observed under any of these conditions. In line with this finding, treatment of cells with PROTACs did not affect the ubiquitylation status of RNF4 (Fig. 2D). Since the efficiency of PROTACs can be cell-type specific, we validated our compounds in two AML cell lines, NB-4 and OCI-AML2, but again we were unable to detect degradation of RNF4 upon exposure to the CCW16-based degraders at different concentrations and treatment times (Fig. EV2E,F). Since RNF4 is an E3 ubiquitin ligase, possibly ubiquitinating VHL or CRBN after PROTAC addition, we also tested if the synthesized PROTACs induced CRBN or VHL degradation. However, a reduction in CRBN or VHL protein levels was not detected, demonstrating that the PROTACs did not function as RNF4 recruiting molecules (Fig. EV2G). Altogether, these data showed that both the VHL-type and the CRBN-type CCW16-based PROTACS failed to efficiently degrade RNF4. In the following, we therefore aimed to understand the reason for their inefficiency.

## CCW16 and CCW16-based PROTACs induce HMOX1 expression

Based on the lack of CCW16-based protein degradation in our initial PROTAC design, we aimed to comprehensively characterize the cellular properties of CCW16 before designing the next generation of CCW16-based RNF4 degrader molecules. First, we used quantitative mass spectrometry to detect subtle differences in protein levels which may not be quantifiable by Western blotting. We performed TMT-based quantitative proteomics in cells expressing FLAG-tagged RNF4 from a doxycycline-inducible promoter. Cells were either treated with the CRBN-based PROTAC 1a, with the VHL-based PROTAC 2c or with the RNF4 binder CCW16 alone (Figs. 2E and EV2H). In accordance with our immunoblot experiments, no significant reduction of the RNF4 protein level was detected upon treatment of cells with the PROTACs. The most significantly downregulated protein following treatment of cells with either PROTAC 1a or 2c was mitochondrial

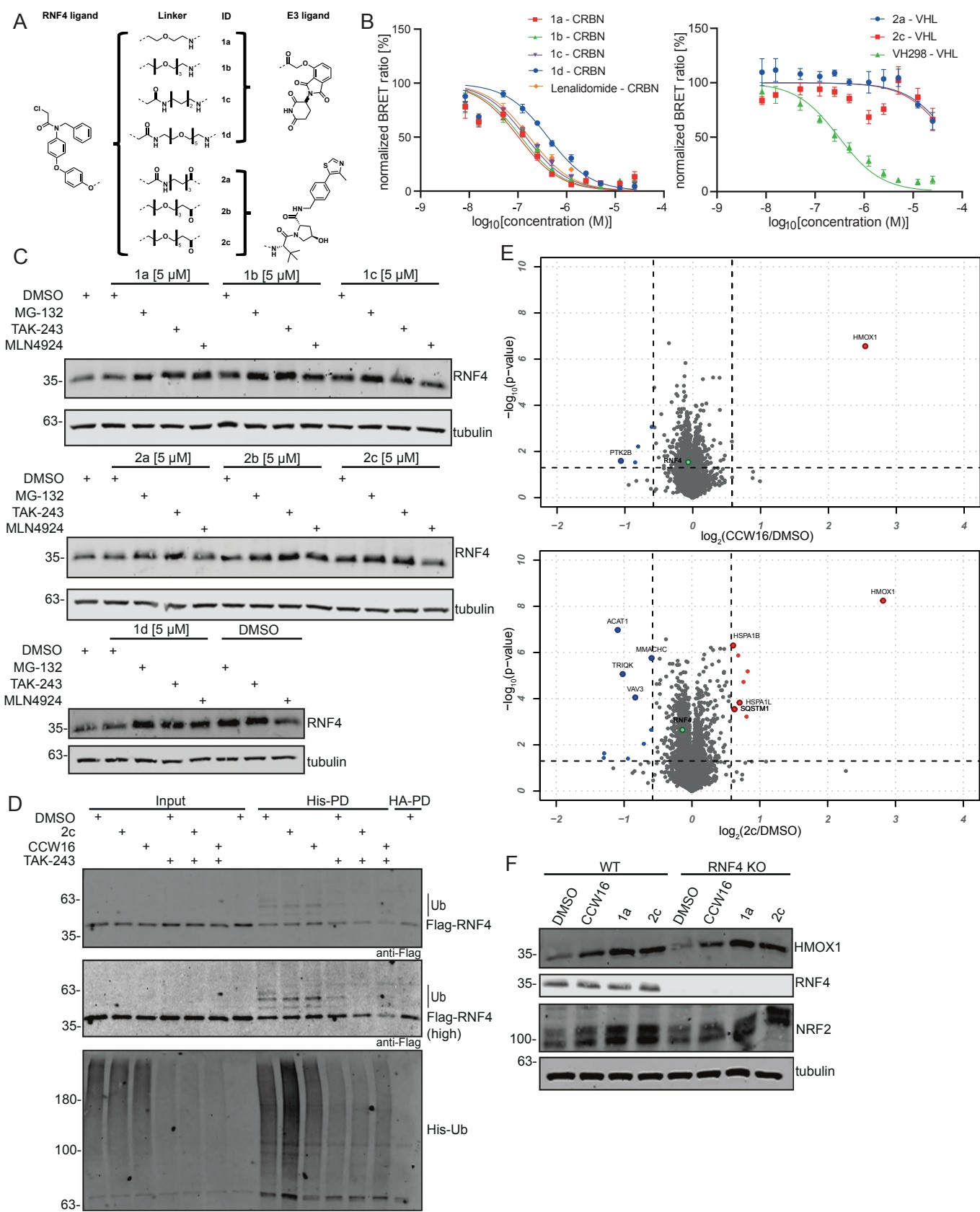

◄ **Figure 2. Design and evaluation of RNF4 targeting PROTACs.**

(A) The RNF4 binder CCW16 (left panel) was utilized for PROTAC development using diverse linkers (central panel) as well as VHL and CRBN E3 ligands (right panel). (B) NanoBRET assay for CRBN and VHL E3 ligand-based RNF4 PROTACs to determine cell membrane permeability in intact cells transiently expressing CRBN (left panel) or VHL (right panel) with an N-terminally tagged NanoLuc. Error bars show the standard deviation of the mean, $n = 4$ (biological replicates). (C) Treatment of HeLa WT cells with different RNF4 targeting PROTACs and evaluation of RNF4 degradation level by immunoblotting. Cells were treated 30 min before PROTAC treatment [5 μM] with MG-132 [20 μM], TAK-243 [1 μM], or MLN4924 [500 nM] and harvested after 6 h. Control cells were treated with DMSO. Tubulin was used as loading control. Same experiment is also shown in Fig. EV2G. (D) HeLa Flag-RNF4 endogenously tagged cells were transfected with His-Ub for 48 h and pretreated with TAK-243 [1 μM], followed by PROTAC **2c** [5 μM] or CCW16 [5 μM] treatment for 6 h. Enrichment of ubiquitylated proteins was performed by denaturing Ni-NTA pulldown. Higher molecular weight bands show ubiquitylation signal. Control cells were treated with DMSO. HA-pulldown was performed as a negative control. (E) Whole cell proteome analysis by mass spectrometry. HeLa cells expressing RNF4 from a doxycycline-inducible promoter were treated either with CCW16 [5 μM] (upper panel) or **2c** [5 μM] (lower panel) for 6 h. Results of the TMT-based MS analysis are visualized in a volcano plot comparing PROTAC treatment vs. DMSO control. Hits considered as significantly upregulated are highlighted in red (log$_2$(ratio) ≥ 0.58, −log$_{10}$(p value) ≥ 1.3) and hits considered as significantly downregulated are highlighted in blue (log$_2$(ratio) ≤ −0.58, −log$_{10}$(p value) ≥ 1.3). The identification of those candidates was based on a two-sided Student's t-test analysis comparing the normalized TMT abundances of CCW16/**2c** treatment with DMSO control treatment. Experiments were performed with four biological replicates. (F) Validation of proteomic results by immunoblotting in HeLa WT and HeLa RNF4 KO cells. Same treatment procedure as in (E). Tubulin was used as loading control. Source data are available online for this figure.

Acetyl-CoA acetyltransferase (ACAT1), which catalyzes the last step of the mitochondrial beta-oxidation pathway. Strikingly, however, the proteomic analysis also revealed a drastic induction in heme oxygenase-1 (HMOX1) levels, which has an important function in protecting cells from oxidative damage (Poss & Tonegawa, 1997), in cells treated with the CRBN-based PROTAC **1a** or VHL-based PROTAC **2c** as well as in cells treated with CCW16 only (Figs. 2E and EV2H; Dataset EV1). This finding was confirmed by immunoblotting using a specific HMOX1 antibody. Furthermore, anti-HMOX1 immunoblotting in RNF4 KO HeLa cells revealed that the upregulation of HMOX1 upon exposure of cells to CCW16 or the CCW16-based PROTACs was independent of RNF4 (Fig. 2F). Interestingly, MS data also revealed that proteins of the heat shock protein family A (Hsp70) were upregulated after treatment with **1a** and **2c** (Figs. 2E and EV2H), indicating a more general stress response after exposure with CCW16 and its PROTAC derivatives. In line with our observation of HMOX1 and Hsp70 upregulation, protein levels of the transcription factor NRF2 (nuclear factor erythroid-derived 2-related factor 2), which controls expression of HMOX1 and other stress-responsive targets (Reichard et al, 2007), were increased after CCW16 and PROTAC treatment (Fig. 2F). Altogether our proteomic data confirmed that CCW16 and CCW16-based PROTACs did not degrade RNF4, but induced a general RNF4 independent oxidative stress response leading to upregulation of HMOX1 and other heat shock proteins.

## CCW16 targets cysteine residues 51 and 91 of RNF4 in vitro

Next, we aimed to confirm binding of CCW16 to RNF4 in vitro using purified recombinant protein. CCW16 has been reported to covalently target cysteine residues 132 and 135 of RNF4 residing within its RING domain (Ward et al, 2019). To confirm this interaction, we first expressed GST-RNF4 wild-type (WT) and a GST-RNF4 C132/135S mutant in *E. coli* and purified the respective proteins to homogeneity using GSH-beads (Fig. 3A). The purified GST-RNF4 was incubated with biotin-linked CCW16 (biotin-CCW16) using the established CCW16 exit vector for coupling the biotin moiety (Appendix). After stringent washing, samples were separated by SDS-PAGE and biotin-CCW16-RNF4 conjugates were detected using streptavidin linked to a fluorophore (Fig. 3A). Under

these conditions, biotinylation was detected in both wild-type RNF4 and the RNF4 C132/135S variant (Fig. 3B). Importantly, binding of biotin-CCW16 to GST alone was not detected indicating that in vitro CCW16 bound to RNF4, but not via the proposed cysteine residues C132 and C135.

To identify cysteine residues in RNF4 modified by CCW16, we performed a mass spectrometry mapping experiment. To this end, we incubated purified GST-RNF4 WT with CCW16 or the CCW16-based PROTAC **2a** and analyzed the resulting peptides by tandem mass spectrometry after trypsin digestion (Fig. 3C; Dataset EV2). Importantly, we identified by MS/MS a peptide in which CCW16 was conjugated to cysteine 91, as indicated by a mass shift in the peptide containing residue (Figs. 3D (pink) and EV3A), but we were unable to detect conjugation to cysteine residues C132/135 or any other cysteine residue of the RING domain (Figs. 3D, blue and green and EV3B). Notably, when RNF4 was incubated with the CCW16-based PROTAC **2a**, we also observed the formation of a covalent bond between **2a** and RNF4 at cysteine C91 (Figs. 3D (pink) and EV3C). While we did find the cysteines C132/C135 to be modified by PROTAC **2a**, we only detected the modified peptide in one replicate with very low intensity, and we were unable to map the modification to one specific residue, but a doubly modified peptide was not found (Fig. EV3B).

To validate these data, we applied the same approach as described in Fig. 3A, but used an RNF4 variant, where C91 was mutated to a serine residue. When comparing pulldown levels with wild-type RNF4, the C91S mutant exhibited reduced covalent bond formation with CCW16, but the mutant likely still exhibited additional sites of covalent attachment (Fig. 3E). Our mass spectrometry analysis did not cover a long peptide within the unstructured N-terminus of RNF4, which harbors C51 (Fig. 3D, gray). To validate whether C51 served as an additional CCW16 covalent attachment site, we generated an RNF4 variant, in which C51 was mutated to serine as well as a double C51/91S mutant (Fig. 3F). Gratifyingly, no biotinylation was observed using the double mutant demonstrating that biotin-CCW16 covalently bound to C51 and C91 located in the unstructured region of RNF4 (Fig. 3D,F).

## BRD4 degradation via CCW28-3 does not depend on RNF4

Because of the high reactivity of CCW16 required to react with two cysteines in the unstructured protein region of RNF4, we

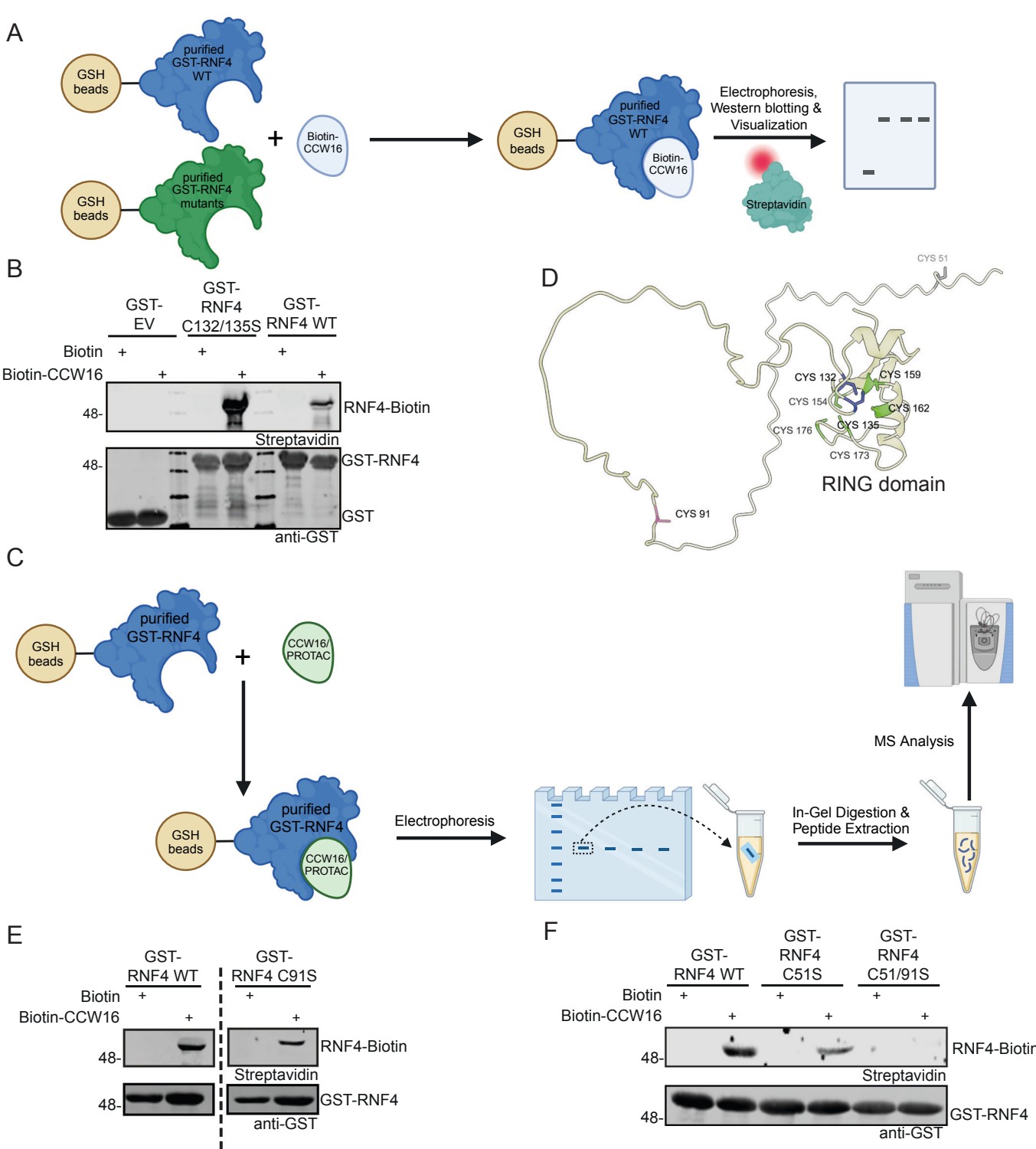

hypothesized that the RNF4-based BRD4 degrader CCW16-JQ1 (CCW28-3) may not degrade BRD4 via an RNF4-dependent mechanism (Ward et al, 2019). To investigate this, we compared CCW28-3 degradation in HeLa WT with degradation in HeLa RNF4 KO cells. The established CRBN ligand-based PROTAC dBET6 served as a positive control. As reported (Ward et al, 2019), BRD4 immunoblotting revealed significant degradation of the short

and the long isoform of BRD4 after CCW28-3 or dBET6 treatment in HeLa WT cells (Fig. EV4A,B). However, similar levels of degradation were also observed in RNF4 KO cells using CCW28-3 (Fig. EV4A,B), suggesting RNF4-independent degradation of BRD4 by CCW28-3. Recovery of BRD4 levels was observed by co-treatment with the proteasome inhibitor MG-132, confirming a proteasome-dependent BRD4 degradation. Interestingly, however,

**Figure 3.  Further evaluation of RNF4-binder CCW16 in vitro.**

(A) Scheme outlining in vitro interaction studies of biotin-CCW16 and GST-based purified RNF4. Biotin-CCW16 modification of RNF4 was visualized by immunoblotting with a streptavidin protein linked to a fluorophore. (B) Biotin or biotinylated CCW16 was incubated with wild-type GST-RNF4 or mutants with cysteine residues changed to serines as indicated. Covalent biotin-CCW16-RNF4 conjugates were detected by streptavidin immunoblotting. (C) Experimental procedure of in vitro interaction studies of CCW16 or PROTAC **2a** with GST-RNF4 WT to identify modified cysteine residues in RNF4 by mass spectrometry. The experiment was performed with three replicates. (D) AlphaFold2 (Jumper et al, 2021; Varadi et al, 2022, 2024) model of human RNF4, highlighting the CCW16 modified cysteine residue identified in the mass spectrometry experiment in (C) in pink, the two catalytic, but unmodified cysteine residues are shown in blue, the not detectable cysteine residue 51 is shown in gray and all remaining cysteine residues are highlighted in green. (E, F) Same procedure as in (B). GST-RNF4 cysteine to serine mutations are indicated. Covalent biotin-CCW16-RNF4 conjugates were detected by streptavidin immunoblotting. Source data are available online for this figure.

treatment of cells with the neddylation inhibitor MLN4924, also restored BRD4 expression (Fig. EV4C). Since MLN4924 only affects cullin-dependent RING E3 ubiquitin ligases (Duda et al, 2008), but not RNF4, this data confirmed that CCW28-3 induced RNF4-independent degradation via a different but so far unidentified cullin-dependent RING E3 ligase.

To further validate the RNF4-independent activity of CCW28-3, we performed an RNF4 KD in HEK293T cells expressing an endogenously HiBiT-tagged BRD4 and treated cells with CCW28-3 or CCW16 alone for 4, 6, and 24 h (Fig. EV4D,E). While CCW16 alone did not affect BRD4 levels, CCW28-3 treatment led to a decrease of HiBiT-tagged BRD4 (Fig. EV4D). However, the BRD4-HiBiT signal was not restored after siRNA-mediated depletion of RNF4, confirming RNF4-independent BRD4 degradation (Fig. EV4E). Efficient KD was confirmed by anti-RNF4 immunoblotting (Fig. EV4F), and viability assays showed that the viability of HEK293T BRD4-HiBiT cells treated with CCW28-3 was only affected at concentrations in the high micromolar range and at treatment times longer than 9 h (Fig. EV4G).

## CCW16 covalently targets a broad spectrum of cellular proteins, including peroxiredoxins

To investigate the selectivity of CCW16 in the cellular environment, cell lysates from HeLa cells were incubated with biotin-CCW16 or with biotin alone, followed by streptavidin pulldown and mass spectrometry analysis (Fig. 4A). In the biotin-CCW16 pulldown, around 1200 proteins were significantly enriched compared to the biotin control pulldown (Fig. EV5A; Dataset EV3). Analysis of the mass spectrometry data and the modified peptides revealed 38 proteins, in which we identified specific cysteine residues which were modified by covalent biotin-CCW16 adducts (Fig. 4B).

Peroxiredoxins (PRDXs) are important for the clearance of peroxides by oxidizing a thiol group of a cysteine residue in PRDX, resulting in the reduction of peroxides (Kim et al, 1988; Li et al, 2020). Six different PRDXs (PRDX1-6) are expressed in mammals. PRDX1, 2 and 6 are mainly localized in the cytosol, while PRDX3 is present in the mitochondria, and PRDX4 is found in the endoplasmic reticulum and is also localized in the extracellular space. PRDX5 is distributed to the cytosol, the mitochondria, the nucleus and the peroxisomes (Li et al, 2020). Intriguingly, three paralogs, PRDX1, PRDX2 and PRDX6 showed covalent binding to biotin-CCW16 on their catalytically active cysteine residues (Figs. 4B,C and EV5A). GOBP analysis of all biotin-CCW16 modified proteins revealed a significant enrichment of proteins involved in cell redox homeostasis (Fig. EV5B). These data suggested that the binding of CCW16 caused inhibition of PRDX activity, resulting in oxidative stress, which was also supported by the observed

upregulation of HMOX1 upon treatment of cells with CCW16 or CCW16-based PROTACs (Figs. 2E and EV2H). In agreement with mass spectrometry data, immunoblotting data confirmed the interaction with a large number of biotin-CCW16 modified proteins compared to the biotin control after biotin-CCW16 pulldown (Fig. 4D). Furthermore, we also confirmed interaction of PRDX1 with biotin-CCW16 by immunoblotting, validating also our mass spectrometry results (Fig. 4D). However, we also observed a very weak band for RNF4 in the pulldown fraction (Fig. EV5C). This data confirms our hypothesis that CCW16 formed covalent bonds in a non-selective manner with accessible cysteine residues via the reactive chloro-*N*-acetamide warhead, thereby inducing redox imbalance and oxidative stress.

## CCW16 treatment induces lipid peroxidation followed by ferroptosis in AML cells

Upregulation of HMOX1 is a hallmark of the ferroptosis cell death pathway (Wu et al, 2022). Furthermore, Kyoto Encyclopedia of Genes and Genomes (KEGG) analysis of our proteomic data revealed a significant enrichment of proteins associated with the ferroptosis pathway (Figs. 2E and EV5D). Since ferroptosis is driven by lipid peroxidation (Stockwell et al, 2017; Reichert et al, 2020; Su et al, 2020b; Dixon and Olzmann, 2024), we investigated whether our compounds led to lipid peroxidation in AML cells. We treated OCI-AML2 cells either with CCW16 alone, with the PROTAC **2c** or with cumene hydroperoxide (HP) as a positive control for 2.5 and 2 h, respectively. To analyze lipid peroxidation, cells were treated for the last 30 min with the fluorescent conjugated fatty acid dye C11-BODIPY581/591, which changes color from red to green fluorescence upon oxidation after accumulation in membranes, and flow cytometry was performed (Pap et al, 1999). We detected a highly significant induction of lipid peroxidation after PROTAC **2c** treatment as well as after treatment with CCW16 alone, which was comparable with peroxidation levels after HP treatment, suggesting that CCW16 and PROTAC **2c** were involved in the activation of lipid peroxidation (Fig. 5A).

To evaluate if the lipid peroxidation resulted in the induction of ferroptosis, we pretreated the OCI-AML2 cells with ferrostatin-1, a ferroptosis inhibitor (Miotto et al, 2020), followed by treatment with CCW16 alone or with all our synthesized CRBN- and VHL-based PROTACs and measured the effect on cell viability via a CTG assay (Fig. 5B,C). The GPX4 inhibitor RSL3, a well-known ferroptosis inducer leading to a significant reactive oxygen species (ROS) accumulation, was used as a positive control (Yang et al, 2014). Analysis revealed that most of the PROTACs led to a strong reduction in cell viability, which was rescued by ferroptosis inhibition (Fig. 5B,C). Within the set of tested PROTACs, a trend

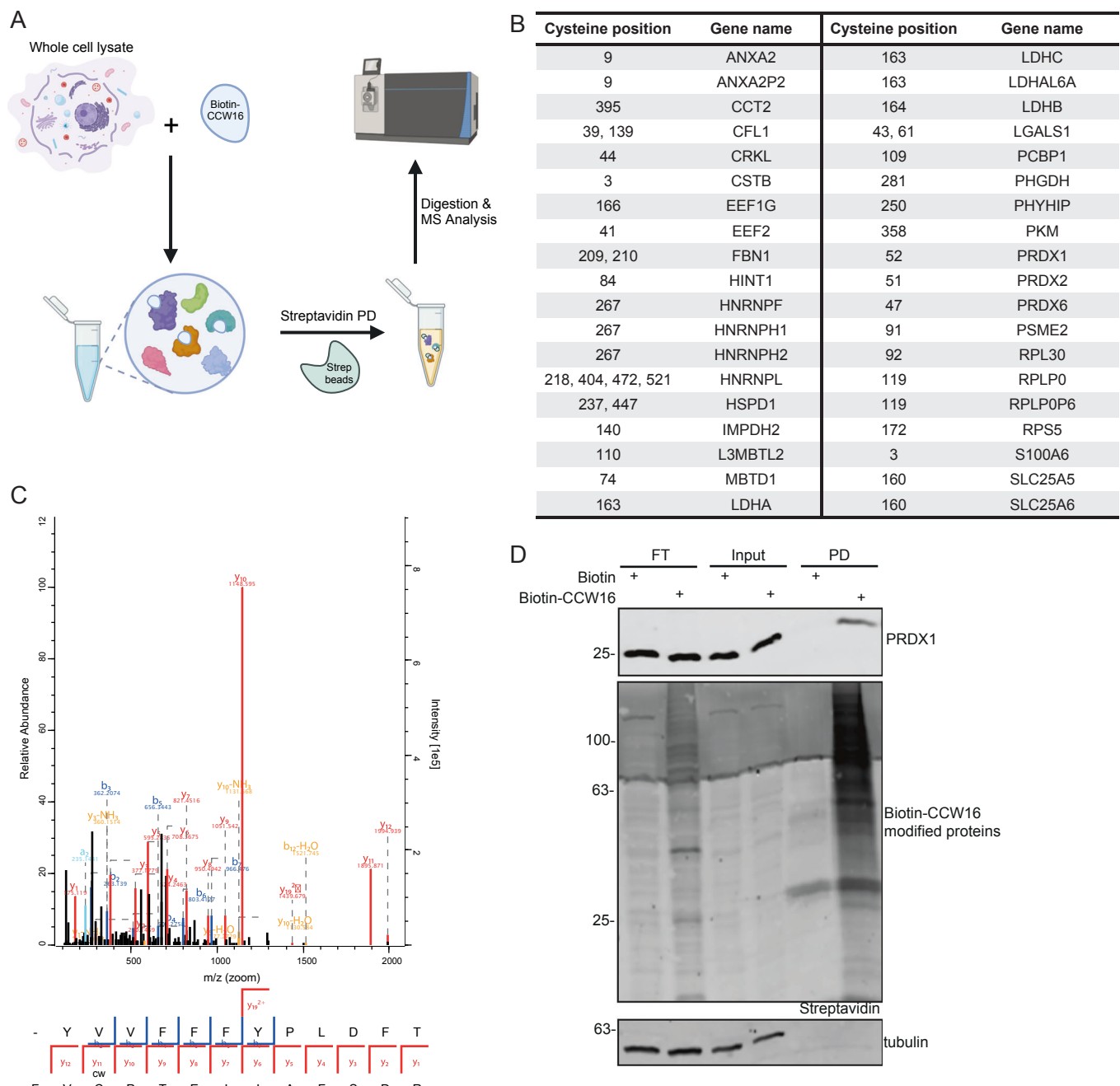

**Figure 4. Identification of CCW16 targets in a cellular system.**

(A) Experimental scheme to identify cellular targets of CCW16. Whole HeLa WT cell lysate was incubated with biotin [10 μM] or biotin-CCW16 [10 μM], followed by streptavidin pulldown to enrich for biotin-CCW16 modified proteins, on-bead digestion and quantitative MS analysis. (B) All detectable proteins from the mass spectrometry experiment in (A) which were modified by biotin-CCW16. Gene names and respective modified cysteine residues are indicated. CCW16 target proteins were defined by two-sided Student's *t*-test analysis comparing LFQ intensities of biotin-CCW16 pulldown with the respective biotin control pulldown. Experiments were performed as biological triplicates. (C) MS Spectrum of biotin-CCW16 modified PRDX1. (D) Validation of streptavidin pulldown results by immunoblotting in HeLa WT cells. The same treatment procedure as in (A). Tubulin was used as loading control. FT flow through; PD pulldown. Source data are available online for this figure.

towards reduced effects on cell viability was observed for PROTACs carrying an amide group connecting the CCW16 ligand and the chemical linker moiety, which was introduced for synthetic reasons, compared to the derivatives and CCW16 lacking an amide bond at this position. Furthermore, cell viability correlated with the cellular permeability of the compounds as estimated by the NanoBRET™. The RNF4 binder alone also led to a significant decrease in cell viability (Fig. 5B,C), indicating that CCW16 is the chemical moiety triggering the induction of lipid peroxidation followed by ferroptosis.

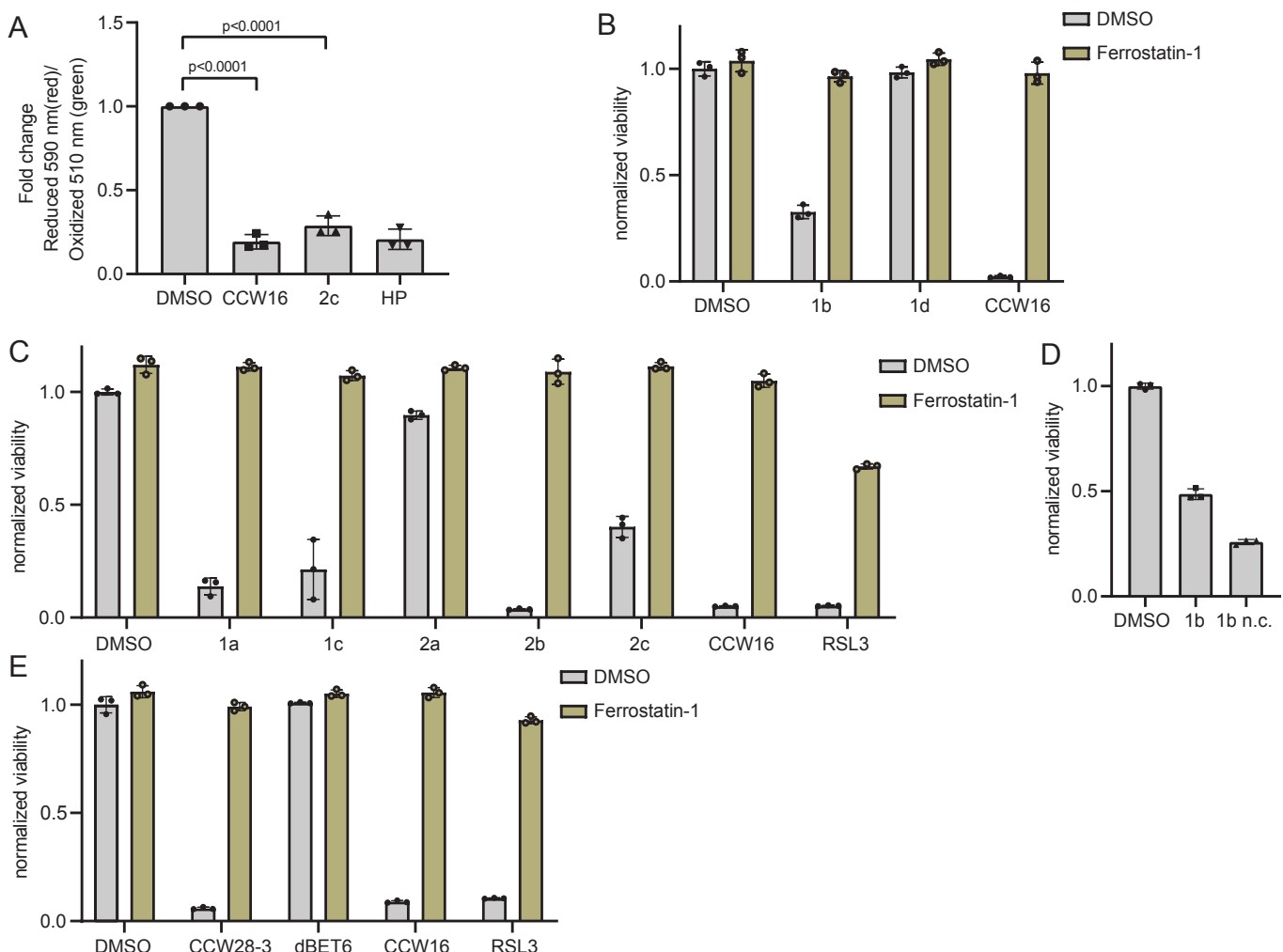

**Figure 5. Analysis of the treatment effects of RNF4 binder CCW16 in a cellular system.**

(A) Treatment of OCI-AML2 cells for 2.5 h with CCW16/2c [each 2.5 µM] or for 2 h with cumene hydroperoxide [100 µM] (HP, positive control) followed by labeling with BODIPY® 581/591 C11 fluorophor to measure lipid peroxidation. Upon oxidation in living cells, fluorescence shifts from red (reduced) to green (oxidized). The fluorescence signal was measured by flow cytometry (events: 10,000 cells). Ratio of reduced vs. oxidized signal is indicated. *P* values of two-tailed, unpaired Student's *t*-tests are indicated; error bars show the standard deviation of the mean. The experiment was performed in three biological replicates. CCW16: *p* < 0.0001, 2c: *p* < 0.0001. (B, C) Pretreatment of OCI-AML2 cells with 10 µM of ferrostatin-1 1 h before treatment with either 1 µM of **1b, 1 d** or CCW16 (B) and **1a, 1c, 2a, 2b, 2c,** CCW16 or RSL3 (positive control, 10 µM) (C) for 6 h and measurement of viability with the CellTiterGlo assay. Error bars show the standard deviation of the mean, *n* = 3 (biological replicates). (D) Treatment of OCI-AML2 with either 1 µM of PROTAC **1b** or the inactive counterpart **1b n.c.** for 6 h. Measurement of viability by CellTiterGlo. Error bars show the standard deviation of the mean, *n* = 3 (biological replicates). (E) Pretreatment of OCI-AML2 with ferrostatin-1 for 1 h [10 µM], followed by treatment with CCW28-3 [1 µM], dBET6 [500 nM], CCW16 [1 µM], or RSL3 [10 µM] for 6 h and measurement of cell viability by CellTiterGlo. Error bars show the standard deviation of the mean, *n* = 3 (biological replicates). Source data are available online for this figure.

To exclude that interaction with CRBN was involved in ferroptosis induction, we synthesized a negative control of PROTAC **1b** (**1b n.c.**). **1b n.c.** was identical in its chemical structure to **1b**, but contained an *N*-methyl thalidomide instead of thalidomide, preventing CRBN binding. However, also **1b n.c.** strongly decreased viability, confirming that ferroptosis was induced by the electrophile and not through CRBN interaction of this PROTAC series (Fig. 5D). Furthermore, we investigated the viability of OCI-AML2 cells after treatment with the RNF4-based BRD4 degrader CCW28-3 and compared it with the treatment of the CRBN-based BRD4 degrader dBET6. CCW28-3 treatment induced a significant decrease in cell viability, which was rescued by ferroptosis inhibition (Fig. 5E). By contrast, dBET6 did not

influence the survival of OCI-AML2 cells (Fig. 5E). Notably, the degradation of BRD4 by CCW28-3 was not abrogated by ferrostatin-1, indicating that it is not linked to the ferroptotic activity of CCW28-3 (Fig. EV5E). In conclusion, our data revealed that CCW16 covalently bound to a large variety of different proteins, thereby inducing a cell death mechanism based on oxidative stress and ferroptosis in an RNF4-independent manner.

## Chloro-*N*-acetamide ligands exhibit ferroptotic activity in a GPX4-sensitive pathway

CCW16 comprises a chloro-*N*-acetamide group as the reactive electrophile targeting cysteine residues (Ward et al, 2019). To test

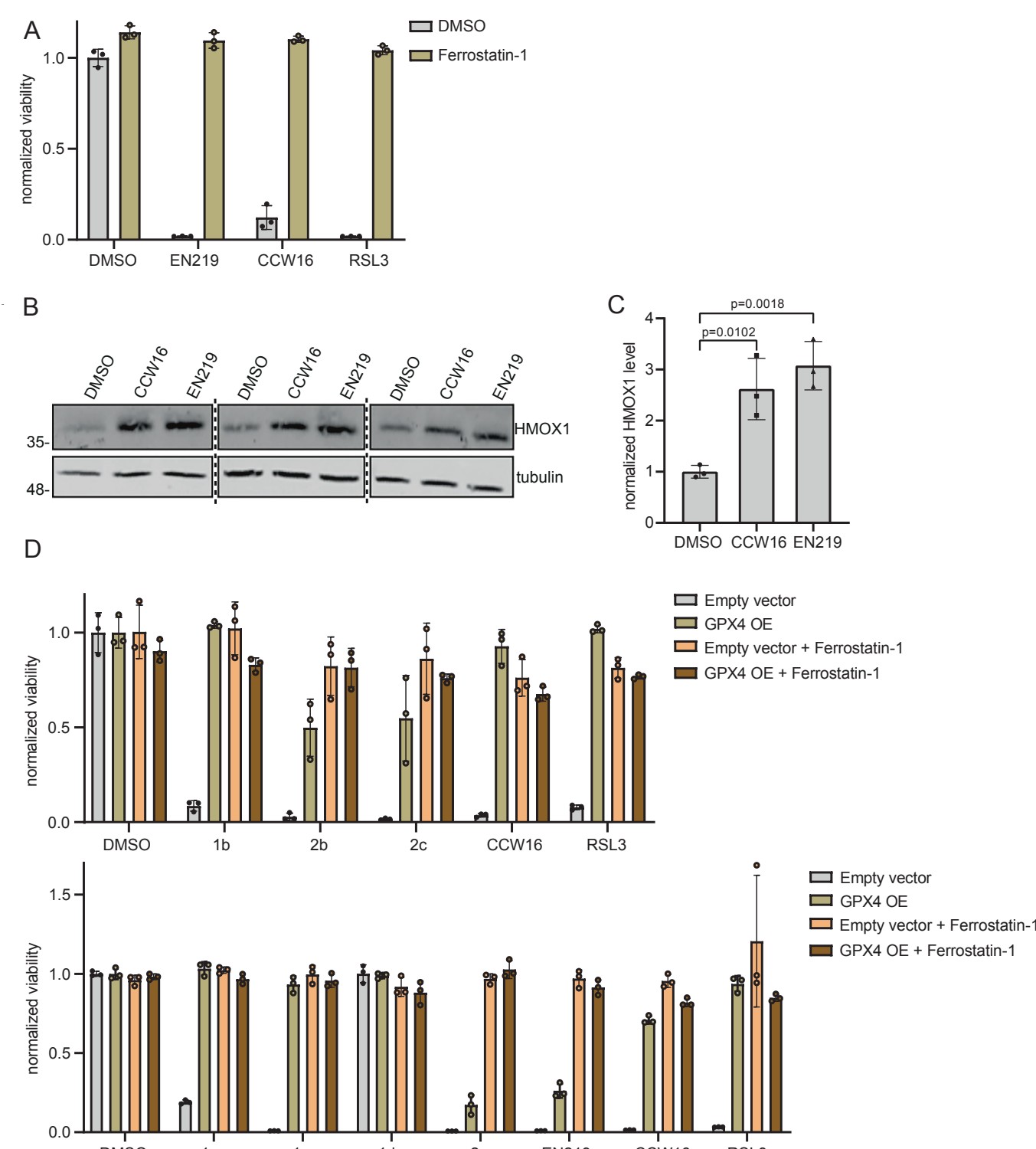

whether other chloro-*N*-acetamide-containing compounds generally exert a ferroptotic potential, we included the covalent ligand EN219 in our analysis, which was reported to target the E3 RING-type ligase RNF114 (Luo et al, 2021). Strikingly, as described for CCW16, EN219 strongly induced ferroptosis that was blocked by ferrostatin-1 (Fig. 6A). Furthermore, similar to CCW16, EN219 also strongly upregulated HMOX1 expression in HeLa cells, indicating that EN219

triggered oxidative stress in cells (Fig. 6B,C). GPX4 is a critical regulator of ferroptosis and a key target of RSL3. Accordingly, overexpression of GPX4 is expected to rescue RSL3-induced ferroptosis (Sui et al, 2018). To confirm this mechanism, we wanted to determine whether GPX4 expression could rescue CCW16- and EN219-mediated ferroptotic cell death. We monitored the cell viability of parental HT-1080 cells and HT-1080 cells stably overexpressing

**Figure 6.  Analysis of different chloro-*N*-acetamide ligands in a cellular system.**

(A) Pretreatment of OCI-AML2 with ferrostatin-1 for 1 h [10 µM], followed by treatment with EN219 [1 µM], CCW16 [1 µM], or RSL3 [10 µM] for 6 h and measurement of cell viability by CellTiterGlo. Error bars show the standard deviation of the mean, $n = 3$ (biological replicates). (B) HeLa WT cells were treated either with EN219 [5 µM] or CCW16 [5 µM] for 6 h and analyzed by immunoblotting. The experiment was performed with three biological replicates. Tubulin was used as loading control. (C) Quantification of HMOX1 levels shown in (B) normalized to tubulin loading control and DMSO control treatment. $P$ values of two-tailed, unpaired Student's $t$-tests are indicated; error bars show the standard deviation of the mean, $n = 3$ (biological replicates). CCW16: $p = 0.0102$, EN219: $p = 0.0018$. (D) HT-1080 cells stably overexpressing an empty vector or GPX4 WT (GPX4 OE) were treated for 6 h with CCW16/CCW16-based PROTACs [5 µM] or RSL3 as a positive control [1 µM] after treatment with ferrostatin-1 [10 µM] for 1 h. Cell viability was measured by CellTiterGlo. Error bars show the standard deviation of the mean, $n = 3$ (biological replicates). OE overexpression. Source data are available online for this figure.

GPX4 (Fig. EV5F). Strikingly, RSL3, EN219 and CCW16, as well as CCW16-derived PROTACs, did not induce cell death in cells overexpressing GPX4, indicating that their ferroptotic potential was linked to the inactivation of GPX4 (Fig. 6D).

## Discussion

Data on patient survival and our RNF4 depletion experiments made a compelling case targeting RNF4 for degradation as a new therapeutic strategy for the treatment of AML. Our data show that depletion of the StUbL RNF4 sensitizes cells to decitabine, most likely through impairment of DPC clearance. In line with our data, very recent findings demonstrated that the inhibition of the E3 ubiquitin ligase TOPORS, which acts in parallel with RNF4 in proteasomal clearance of DNMT1-DPCs (Liu et al, 2024; Carnie et al, 2024), also augments the efficacy of hypomethylating agents (Kaito et al, 2024; Truong et al, 2024).

RNF4 does not contain a druggable domain, making it a challenging target for the development of conventional small-molecule inhibitors. However, covalent inhibitors have been developed targeting cysteine residues in RNF4 for covalent bond formation. Here, we aimed to exploit developed ligands for the design of PROTACs using established CRBN- or VHL-ligands and the irreversible RNF4 ligand CCW16 targeting RNF4 for degradation. However, after synthesizing and testing a set of PROTACs, this strategy was not successful. Detailed analysis using recombinant full-length RNF4 as well as cellular extracts revealed covalent binding of CCW16 to two cysteines in RNF4 (C51 and C91), but also covalent attachment to many non-specific covalent interactions in cell extracts including also RNF4. In our proteomic analysis, more than 1200 proteins were captured by biotinylated CCW16 in a streptavidin pulldown experiment. NMR spectra of RNF4, as well as AlphaFold2 models, revealed that RNF4 is largely unstructured except for its RING domain (Murphy et al, 2020). Therefore, cysteines located in the N-terminus of RNF4 are likely to be solvent-exposed and thus easily targeted by covalent inhibitors. However, due to the unstructured nature of RNF4, compounds harboring electrophiles for covalent bond formation have no significant affinity for such disordered targets. Thus, covalent ligands do not accumulate at a specific binding site, which is a commonly used structure-based mechanism leading to the high selectivity of developed covalent enzyme inhibitors (Attwood et al, 2021; Chaikuad et al, 2018). Consequently, electrophiles used for targeting disordered proteins must be highly reactive, as non-covalent interactions with an unstructured binding surface are likely to be weak and cannot enhance the local concentration of covalent compounds. However, transient structural changes may

allow weak association even with intrinsically disordered proteins, suggesting that milder electrophiles may also be used to fine-tune the reactivity and selectivity of covalent ligands.

The chloro-*N*-acetamide electrophile has been frequently used in the development of covalent ligands. For example, chloro-*N*-acetamide is the covalent warhead in the DDB1 ligand MM-02-57 (Meyers et al, 2024), in the RNF114 ligand EN219 (Luo et al, 2021), in a minimal recruitment moiety, targeting DCAF16 (Zhang et al, 2019), as well as in covalent FEM1B ligands (Henning et al, 2022) and many others. Our data suggest that death pathways need to be carefully evaluated to rule out off-target mechanisms leading to the degradation of proteins targeted by PROTACs based on these novel ligands. We demonstrated that well-established tool compounds used for PROTAC validation, such as proteasome inhibitors, can produce inconclusive results when used at inappropriate concentrations that cause toxicity (Schwalm et al, 2025). We recently proposed a workflow for evaluating the suitability of E3 ligands for PROTAC development that specifically monitors cellular toxicity levels of PROTACs (Miletić et al, 2025). Highly reactive ligands, such as the approved drugs used for the treatment of multiple sclerosis, have been described to covalently bind to cysteines within the BTB domain of the E3 ligase Keap1 (Xu et al, 2015). However, these drugs also induce oxidative stress in cells, resulting in the induction of the expression of antioxidative transcription factors such as NRF2. Here, we also observed induction of oxidative stress and the cell death program ferroptosis in cells treated with chloro-*N*-acetamide-containing PROTACs. This observation prompted us to reexamine published CCW28-3–based degraders targeting BRD4 (Ward et al, 2019). However, our data suggested that RNF4-based BRD4 degrader CCW28-3 did not function through an RNF4-dependent pathway, as BRD4 degradation was observed in RNF4-deficient cells. Although we cannot completely rule out that BRD4 degradation is mediated by residual RNF4 present in our cellular models, the impairment of BRD4 degradation by the neddylation inhibitor MLN4924 provided further evidence that CCW28-3-mediated BRD4 degradation was RNF4 independent and required a yet to be identified cullin-based E3 ligase.

In this study, we propose that the high reactivity of the chloro-*N*-acetamide results in the stress signature observed in our proteomic study, which revealed a dramatic upregulation of HMOX1 together with enhanced expression of NRF2. NRF2 is a key transcription factor in redox signaling and an inducer of cytoprotective key enzymes, including HMOX1, an important protein for the protection of cells from oxidative damage during stress by degrading heme (Abraham and Kappas, 2008). Furthermore, HMOX1 is a key marker of ferroptosis, a specific type of programmed cell death characterized by lipid peroxidation caused by enhanced ROS production and iron overload (Dixon et al, 2012). We observed a strong ferroptotic potential of CCW16 and the CCW16-based

PROTAC molecules in AML cells, which are known to be very sensitive to this pathway (Ghaffari, 2008). Importantly, our data demonstrated that the ferroptotic activity of CCW16 was independent of RNF4 or its linkage to a CRBN- or VHL-binding ligand. This finding further supports our hypothesis that CCW16 has intrinsic ferroptotic potential that can be abrogated by the ferroptosis inhibitor ferrostatin-1. Lipid peroxidation is primarily detoxified by the action of glutathione peroxidase 4 (GPX4) or peroxiredoxins, such as PRDX6 (Yang et al, 2014; Lu et al, 2019; Chen et al, 2024). Interestingly, we identified the active cysteine residues of PRDX1 (C52), PRDX2 (C51), and PRDX6 (C47) as direct covalent interaction sites of CCW16, suggesting that inhibition of these PRDX proteins by CCW16 contributed to the observed ferroptosis phenotype. High expression of PRDX1 has been correlated with an unfavorable prognosis in AML patients, indicating that targeting of PRDX1 by CCW16 might be a therapeutic option in a subset of AML patients (Hassan et al, 2024). Notably, by using a CCW16 derivative that contained an alkyne handle, the Nomura group demonstrated targeting of HMOX2 and GPX4 by CCW16. These data supported the suggestion that CCW16 preferentially interacts with highly reactive cysteines present in proteins involved in redox signaling (Ward et al, 2019). Importantly, our data also revealed a strong ferroptotic activity of EN219, which targets an intrinsically disordered region in the RING-type ligase RNF114 via a chloro-*N*-acetamide electrophile. Notably, an alkyne-functionalized EN219 probe used for biotin-streptavidin enrichment also bound to GPX4, indicating that GPX4 is a preferred target of chloro-*N*-acetamide electrophiles (Luo et al, 2021). In line with this hypothesis, we showed that overexpression of GPX4 fully rescued ferroptotic cell death induced by EN219, CCW16 or CCW16-derived PROTACs. Interestingly, the small molecule RSL3 that is used as a covalent inhibitor of GPX4 (Yang and Stockwell, 2008) also contains a chloro-*N*-acetamide. Although it has been demonstrated that RSL3 shows low proteome-wide selectivity (Eaton et al, 2020), RSL3-induced ferroptotic cell death can also be rescued by overexpression of GPX4.

Based on its selectivity, CCW16 is unlikely to represent a suitable lead compound for translational studies. However, this compound may have potential as an experimental inducer of ferroptosis in cellular systems. Whether the ferroptotic activity of CCW16 and other chloro-*N*-acetamide-containing compounds could also be exploited therapeutically remains to be determined.

# Methods

### Reagents and tools table

| Reagent/resource | Reference or source | Identifier or catalog number |
|---|---|---|
| **Experimental models** | | |
| HEK293T (*H. sapiens*) | ATCC (CRL-3216) | RRID: CVCL_0063 |
| HEK293T BRD4-HiBiT (*H. sapiens*) | Gifted by Promega | N/A |
| HeLa (*H. sapiens*) | ATCC (CCL-2) | RRID: CVCL_0030 |
| HeLa RNF4-Flag (*H. sapiens*) | Provided by J. Keiten-Schmitz | N/A |
| HeLa RNF4 KO (*H. sapiens*) | Provided by J. Keiten-Schmitz | N/A |
| HeLa Trex Flp in RNF4 WT (*H. sapiens*) | Provided by J. Keiten-Schmitz | N/A |

| Reagent/resource | Reference or source | Identifier or catalog number |
|---|---|---|
| MV4-11 Cas9 (*H. sapiens*) | Gifted by F. Bassermann | N/A |
| NB-4 (*H. sapiens*) | ACC 207 (DSMZ) | N/A |
| OCI-AML2 (*H. sapiens*) | Leibniz-Institut DSMZ | RRID: CVCL_1619 |
| OCI-AML3 Cas9 (*H. sapiens*) | Gifted by H. Uckelmann | N/A |
| **Recombinant DNA** | | |
| pCMV HA-ubiquitin | (Treier et al, 1994) | N/A |
| pCMV His-ubiquitin | (Treier et al, 1994) | N/A |
| pGEX4-T1 | GE-Healthcare | 28-9545-49 |
| pGEX4-T1-RNF4 WT | This study | N/A |
| pGEX4-T1-RNF4 C132/125S | This study | N/A |
| pGEX4-T1-RNF4 C91S | This study | N/A |
| pGEX4-T1-RNF4 C51S | This study | N/A |
| pGEX4-T1-RNF4 C51/91S | This study | N/A |
| pUSEPR-IS_Improved scaffold (empty) | Provided by H. Uckelmann | N/A |
| pUSEPR-IS_sgRNA NTC_1 | Provided by H. Uckelmann | N/A |
| pUSEPR-IS_sgRNA NTC_2 | Provided by H. Uckelmann | N/A |
| pUSEPR-IS_sgRNA RPA3_1 | Provided by H. Uckelmann | N/A |
| pUSEPR-IS_sgRNA RPA3_2 | Provided by H. Uckelmann | N/A |
| pUSEPR-IS_sgRNA RNF4_1 | This study | N/A |
| pUSEPR-IS_sgRNA RNF4_2 | This study | N/A |
| pUSEPR-IS_sgRNA RNF4_3 | This study | N/A |
| **Antibodies** | | |
| anti-BRD4 (rabbit), monoclonal, 1:1000 | Abcam | Cat# ab128874; RRID: AB_11145462 |
| anti-CRBN (rabbit), polyclonal, 1:1000 | Invitrogen | Cat# PA5-38037; RRID: AB_2554641 |
| anti-Flag (M2) (mouse) 1:1000 | Sigma | Cat# F1804; RRID: AB_262044 |
| anti-GPX4 (rabbit), monoclonal, 1:1000 | Abcam | Cat# ab125066; RRID: AB_262044 |
| anti-GST (B14) (mouse), monoclonal, 1:1000 | Santa Cruz | Cat# sc-138; RRID: AB_627677 |
| anti-His Tag (mouse), monoclonal, 1:1000 | Santa Cruz | Cat# sc-53073; RRID: AB_783791 |
| anti-HMOX1 (rabbit), polyclonal, 1:1000 | Proteintech | Cat# 10701-1-AP; RRID: AB_2118685 |
| anti-mouse IgG-IRDye 800CW, polyclonal, 1:10,000 | Li-COR | Cat# 926-32210; RRID: AB_621842 |
| anti-NRF2 (rabbit), polyclonal, 1:5000 | Proteintech | Cat# 16396-1-AP; AB_2782956 |
| anti-PRDX1 (rabbit), polyclonal, 1:5000 | Proteintech | Cat# 15816-1-AP; RRID: AB_2170318 |
| anti-rabbit IgG-IRDye 800CW, polyclonal, 1:10,000 | Li-COR | Cat# 926-32211; RRID: AB_621843 |

| Reagent/resource | Reference or source | Identifier or catalog number |
|---|---|---|
| anti-RNF4 (rabbit), polyclonal, 1:2000 | Proteintech | Cat# 17810-1-AP; RRID: AB_2878443 |
| anti-RNF4 (rabbit), 1:500 | Gift from Vertegaal group | FI-5 |
| anti-SENP6 (rabbit), polyclonal, 1:6000 | Sigma | Cat# HPA024376; RRID: AB_1856678 |
| anti-VHL (rabbit), polyclonal, 1:1000 | Cell Signaling | Cat# 68547S; RRID: AB_2716279 |
| anti-vinculin (mouse), monoclonal, 1:10,000 | Sigma-Aldrich | Cat# V9131; RRID: AB_477628 |
| anti-β-tubulin (E7) (mouse), 1:3000 | Developmental studies hybridoma bank | clone E7, RRID: AB_2315513 |
| **Oligonucleotides and other sequence-based reagents** | | |
| MISSION® siRNA Universal Negative Control #1 | Merck | SIC001 |
| sigl2: 5′ (CGUACGCGGAAUACUUCG xA)TT 3′ | Eurofins | N/A |
| siRNF4_UTR 5′ (GGGCAUGA AAGGUUGAGAA)TT 3′ | Eurofins | N/A |
| siSENP6 5′ (GGA CAA AUC UGC UCA GUG U)TT 3′ | Eurofins | N/A |
| sgRNF4_1 5′ (GAGTATCCG TCCATGCAGAT) 3′ | Merck | N/A |
| sgRNF4_1 5′ (ATCTGCATGG ACGGATACTC) 3′ | Merck | N/A |
| sgRNF4_2 5′ (ACAAGATTGT TGACGGTGAG) 3′ | Merck | N/A |
| sgRNF4_2 5′ (CTCACCGTCAA CAATCTTGT) 3′ | Merck | N/A |
| sgRNF4_3 5′ (CTGCATGGAC GGATACTCAG) 3′ | Merck | N/A |
| sgRNF4_3 5′ (CTGAGTATCC GTCCATGCAG) 3′ | Merck | N/A |
| **Chemicals, enzymes and other reagents** | | |
| (2S,4 R)-1-((S)-2-amino-3,3-dimethylbutanoyl)-4-hydroxy-N-(4-(4-methylthiazol-5-yl)benzyl) pyrrolidine-2-carboxamide hydrochloride | Accela | SY279570 |
| (S)-2-(4-(4-chlorophenyl)-2,3,9-trimethyl-6H-thieno[3,2-f][1,2,4]triazolo[4,3-a][1,4] diazepin-6-yl)acetic acid | Ambeed | 202592-23-2 |
| 2,5-dioxopyrrolidin-1-yl 5-((3aS,4S,6aR)-2-oxohexahydro-1H-thieno[3,4-d]imidazol-4-yl) pentanoate | BLD Pharm | BD116733 |
| 2-chloroacetyl chloride | Acros Organics | 147292500 |
| 3-aminopiperidine-2,6-dione hydrochloride | BLD Pharm | bd170886 |
| 4-(4-methoxyphenoxy)aniline | ABCR | AB234626 |
| 4-DMAP | ABCR | AB115737 |
| 4-hydroxyisobenzofuran-1,3-dione | BLD Pharm | bd5674 |
| Acetic acid (glacial) | VWR | 20104.298 |
| Acetonitrile | Sigma-Aldrich | 34851 |

| Reagent/resource | Reference or source | Identifier or catalog number |
|---|---|---|
| Ammonium bicarbonate | Fluka | 9830 |
| Ampicillin, sodium salt | Carl-Roth | K029.3 |
| Aphidicolin | Sigma | A0781-1MG |
| BBr3 (1 M in DCM) | Thermo Scientific | L14880.AE |
| Benzaldehyde | Merck | 801756 |
| Bromphenol blue | Carl-Roth | A512.1 |
| BSA | Carl-Roth | 8076.2 |
| Chloroform | VWRchemical | 67-66-3 |
| cOmplete™, Mini, EDTA-free Protease Inhibitor Cocktail | Roche | 4693159001 |
| Cytarabine | Selleckchem | S1648 |
| Decitabine | MedChemExress | HY-A0004 |
| DIPEA | Carl-Roth | 4105.1 |
| DMEM | Gibco | 41966-052 |
| DMSO | Applichem | A3672 |
| Doxycycline | Fluka | 44577 |
| DTT | Carl-Roth | 6908.2 |
| EDTA | Honeywell/Fluka | 34549 |
| EPPS | Sigma | E9502 |
| Ferrostatin-1 | Merck | SML0583-5MG |
| Fetal bovine serum | Sigma | F7524 |
| FuGENE 4 K | Promega | E5911 |
| Glycerol 99% p. a | Carl-Roth | 7530.4 |
| HATU | Carl-Roth | 2131.2 |
| Hydroxylamine | Sigma | 438227 |
| InstantBlue® Coomassie Protein Stain | Abcam | ab119211 |
| IRDye® 800CW Streptavidin | Li-COR | Cat# 926-32230 |
| Isopropanol | Roth | CP41.1 |
| Isopropyl b-D-1-thiogalactopyranoside (IPTG) | Carl-Roth | 2316.5 |
| Lipofectamine RNAiMAX Transfection Reagent | Thermo Fisher Scientific | 13778150 |
| Lipofectamine2000 | Thermo Fisher Scientific | 11668027 |
| Lysyl endopeptidase (LysC) | Wako | 129–02541 |
| Magnesium chloride | Carl-Roth | 2189.2 |
| Methanol | Roth | AE71.1 |
| Methyliodide (2 M in MTBE) | Sigma-Aldrich | 456756-100 ML |
| MG-132 | Selleckchem | S2619 |
| MLN4942 | MedChemExress | HY-70062 |
| NaCl | Sigma | 31434-5KG-R |
| NP-40 | Applichem | A1694.0250 |
| Opti-MEM | Gibco | 31985-062 |
| Pierce™ Streptavidin Magnetic Beads | Thermo Fisher Scientific | 88816 |
| PMSF | Carl-Roth | 6367.1 |
| Potassium acetate | ABCR | AB118624 |
| Potassium carbonate | VWR | 26,724,291 |
| Protino glutathione agarose 4B beads | Machery-Nagel | 64-17-5 |

| Reagent/resource | Reference or source | Identifier or catalog number |
|---|---|---|
| RPMI1640 | GIBCO | 11530586 |
| SDS | Carl-Roth | CN30.3 |
| Sequencing grade modified trypsin | Promega | V511C |
| Sodium bicarbonate | Carl-Roth | 8551.2 |
| Sodium deoxycholate | Sigma | 30970 |
| Sodium hydride (60% on mineral oil) | Sigma-Aldrich | 452912-5 G |
| Sodium triactoxyborohydride | Fluorochem | 044864 |
| TAK-243 | MedChemExress | HY-100487 |
| TCEP | Sigma | 646547 |
| tert-butyl (17-amino-3,6,9,12,15-pentaoxaheptadecyl) carbamate | Broadpharm | BP-20592 |
| tert-butyl (2-(2-(2-(2-bromoethoxy)ethoxy)ethoxy) ethyl)carbamate | Broadpharm | BP-22233 |
| tert-butyl (2-(2-bromoethoxy) ethyl)carbamate | BLD Pharm | BD304712 95% |
| tert-butyl (4-bromobutyl) carbamate | BLD Pharm | BD91061 |
| tert-Butyl (5-aminopentyl) carbamate | BLD Pharm | BD12434 |
| tert-butyl 1-hydroxy-3,6,9,12,15-pentaoxaoctadecan-18-oate | PurePEG | 433805 |
| tert-butyl 3-(2-(2-(2-hydroxyethoxy)ethoxy)ethoxy) propanoate | PurePEG | 433803 |
| tert-butyl 7-aminoheptanoate | Acrotein | AS00687 |
| tert-Butyl bromoacetate | Alfa Aesar | A14917 |
| TFA | Carl-Roth | P088.1 |
| TMTproTM 16plex reagent | Thermo Fisher Scientific | A44520 |
| Tosyl chloride | ABCR | AB119196 |
| Triethylamine | Sigma-Aldrich | 471283-100 ML |
| Tris-(2-carboxyethyl)-phosphine hydrochloride (TCEP) | Carl-Roth | HN95.3 |
| Tris-(hydroxymethyl)-aminomethane (TRIS) | Carl-Roth | 4855.3 |
| Triton X-100 | Carl-Roth | 3051.2 |
| Trypsin EDTA solution | Pan Biotech | P10-023100 |
| Tween20 | Roth | 9127.2 |
| Urea | Applichem | A1049.1000 |
| **Software** | | |
| AlphaFold2.0 | (Jumper et al, 2021; Evans et al, 2022) | https://github.com/deepmind/alphafold |
| FlowJoTM Software, Version 10.4.2 | Becton, Dickinson and Company | https://www.flowjo.com/ |
| GraphPad Prism, version 8 for windows | GraphPad Software, La Jolla California USA | http://www.graphpad.com/ |
| GraphPad Prism, version 10 for windows | GraphPad Software, La Jolla California USA | http://www.graphpad.com/ |
| Image Studio Lite, version 5.2 | LICORbio software | https://www.licor.com/bio/image-studio/ |
| MaxQuant, version 1.6.17.0 | (Cox et al, 2011) | https://www.bio-chem.mpg.de/5111795/maxquant |
| Perseus, version 1.6.15.0 | (Tyanova et al, 2016) | https://www.bio-chem.mpg.de/5111810/perseus |
| Proteome Discoverer 2.4 | Thermo Fisher Scientific | https://www.thermofisher.com/de/de/home/industrial/mass-spectrometry/liquid-chromatography-mass-spectrometry-lc-ms/lc-ms-software/multi-omics-data-analysis/proteome-discoverer-software.html |
| R Studio, version 2023.06.1 | RStudio: Integrated Development for R. RStudio, Inc. | https://www.rstudio.com/ |
| **Other** | | |
| CellTiter-Glo® Luminescent Cell Viability Assay | Promega | G7571 |
| DC™ Protein Assay Kit I | Bio-Rad | 5000111 |
| Image-iT® lipid sensor C11-BODIPY[581/591] | Thermo Fisher Scientific | D3861 |
| Micro BCA™ Protein Assay Kit | Thermo Fisher Scientific | 23235 |
| NanoBRET® Nano-Glo® Detection Systems | Promega | N1661 |
| Neon™ Transfection System | Invitrogen | MPK10096 |
| Nano-Glo® HiBiT Lytic Detection System | Promega | N3030 |
| Pierce™ BCA™ Protein Assay | Thermo Fisher Scientific | 23225 |

## Cell culture and transfection

HeLa (female, ATCC®, CCL-2™) and HEK293T (female, ATCC® CRL-1573™), purchased from ATCC, and OCI-AML2 (male, Leibniz-Institut DSMZ) cells were cultured in Dulbecco's Modified Eagle Medium (DMEM) (Thermo Fisher Scientific) supplemented with 10% fetal calf serum (Thermo Fisher Scientific), 100 U/ml penicillin, and 100 U/ml streptomycin (Thermo Fisher Scientific) at 37 °C and 5% $CO_2$. HeLa RNF4 KO and HeLa Flp in T-Rex™ RNF4 WT were generated and provided by Dr. Jan Keiten-Schmitz. OCI-AML3 Cas9 and MV4-11 Cas9 were provided by Dr. Hannah Uckelmann. HEK293T BRD4-HiBiT cells were gifted by PRO-MEGA to Prof. Dr. Stefan Knapp. For the transfection of plasmids in adherent cells, the FuGENE transfection reagent was used for 48 h. For siRNA transfection of adherent cells, the Lipofectamine

RNAiMAX reagent (Thermo Fisher Scientific) was used for 72 h. Suspension cells were transfected with siRNA by electroporation by means of the Neon™ Transfection System (Invitrogen) for 72 h, according to the manufacturer´s manual (settings: 1350 V, 35 ms, 1 pulse). MG-132 [20 µM], MLN4924 [500 nM], and TAK-243 [1 µM] were added to cell culture media 30 min prior to PROTAC treatment and incubated for 6 h. Ferrostatin-1 was used with a final concentration of 10 µM and was always pretreated 1 h before PROTAC treatment. DNA-damaging compounds decitabine [1 µM] and aphidicolin [0.075 µg/ml] were given to the cells for 72 h.

## Cellular viability measurements

siRNA transfection in OCI-AML2 was performed as above. After 72 h of knockdown in a six-well plate, 5000 cells per well in 100 µl media were transferred to a white 96-well plate (Thermo Fisher Scientific, 136101) and treated immediately with different antileukemic and DNA damage inducing compounds for further 72 h at 37 °C and 5% $CO_2$. Finally, cell viability was measured with the CellTiter-Glo® Luminescent Cell Viability Assay (Promega, G7570). For this, the plate was equilibrated to room temperature for around 30 min, 100 µl assay reagent was added to each well, incubated for 2 min on an orbital shaker and for 10 min without shaking at room temperature. Filtered luminescence was measured on a plate reader (SYNERGY H1, microplate reader, BioTek), and data were analyzed by GraphPad Prism 8.

Furthermore, 5000 OCI-AML2 cells per well were seeded in 100 µl of media in a white 96-well plate (Thermo Fisher Scientific, 136101) and treated with ferrostatin-1 or control for 1 h at 37 °C and 5% $CO_2$. Afterwards, different PROTACs were added to the plate and incubated for a further 6 h at 37 °C and 5% $CO_2$. Cell viability measurement was performed as above.

Compound toxicity was determined by using the CellTiter-Glo® 2.0 Cell Viability Assay (PROMEGA G9241). For this, 10 µl of HEK293T cells were seeded with 10,000 cells/well into white 384-well plates (Greiner, 784075) and the cells were allowed to attach for 24 h at 37 °C and 5% $CO_2$. After incubation, the compounds were titrated using an Echo acoustic dispenser (Labcyte) and the plate was further incubated for 24 h at 37 °C and 5% $CO_2$. For the assay, 10 µl assay reagent was added and incubated for 10 min at room temperature. Filtered luminescence was measured on a PHERAstar plate reader (BMG Labtech), and data were graphed, using GraphPad Prism 10.

## CRISPR dropout assay

Lentiviruses were produced in HEK293T cells by transfection of plasmids, containing the guideRNA and an RFP gene for the readout, with Lipofectamine™ 2000 transfection reagent. Supernatant, containing the virus, was harvested after 2 days of incubation at 37 °C and 5% $CO_2$. 20,000 cells in 50 µl per well of a 96-well plate (Sarstedt AG & Co. KG. Ref: 83 3924500) of OCI-AML3 and MV4-11, stably expressing Cas9, were transduced on day 0 each with 10 µl of virus after addition of polybrene [1 µg/ml]. Spin infection was performed by centrifugation of the plate for 1.5 h at 37 °C and 1363 g. Afterwards, wells were filled up with media to 200 µl and the plate was incubated at 37 °C and 5% $CO_2$. Transduction efficiency (~30–70%) was measured 3 days after

transduction by flow cytometry (BECKMAN COULTER Model: CytoFLEX S) in a 96-well plate (Corning Incorporated (Costar) Ref: 3897), followed by measuring the viability of transduced cells after 6, 9, 13, and 15 days.

## Western blot analysis

Samples were resuspended in 6x Laemmli buffer and loaded onto an SDS-PAGE for protein separation, followed by transfer to a nitrocellulose membrane with the wet blot method. Before incubation overnight at 4 °C with primary antibody (dissolved in 5% milk in PBS-T, Table EV1) of the respective target, membranes were blocked for 30 min with 5% milk in PBS-T. Membranes were washed four times in PBS-T, followed by addition of secondary antibody (dissolved in 5% milk in PBS-T) for 1 h purchased from LI-COR containing a fluorophore (IRDye® 800CW/680RD goat anti-mouse or goat anti-rabbit) for detection by the Odyssey® DLx Imaging system. Analysis and quantification was performed by means of the Image Studio™ Software from LI-COR.

## BRD4-HiBiT degradation assay

Endogenously BRD4-HiBiT-tagged HEK293T (HEK293T BRD4-HiBiT) cells were obtained as a kind gift from Promega Corp. $6.4 \times 10^5$ cells were seeded in 1.6 ml in a six-well plate, transfected with siRNA as described above and incubated for 72 h at 37 °C and 5% $CO_2$. To measure degradation, 10 µl of a total concentration of $2.5 \times 10^5$ cells/ml in DMEM medium were seeded into white small volume 384-well plates (Greiner, 784075) and allowed to settle overnight. Subsequently, the PROTACs were titrated to the seeded cells, using an Echo acoustic dispenser (Labcyte) and the plate was incubated for the indicated time at 37 °C and 5% $CO_2$. After incubation, HiBiT Lytic detection reagent was prepared by dilution of LgBiT protein (1:100) and lytic substrate (1:50) in Lytic detection buffer (Promega, N3040). For detection, 10 µl of the prepared mix was added to the treated cells and incubated for 10 min at room temperature. The readout was carried out in a PheraStar FSX plate reader (BMG Labtech) using the LUM plus optical module. Degradation data were then plotted with GraphPad Prism 8 software using a normalized three-parameter curve fit with the following equation: $Y = 100/(1 + 10^{(X-LogIC50)})$.

## Ni-NTA pulldown

Pulldown of His-Ub conjugates with Ni-NTA beads (Qiagen) was performed as described previously (Müller et al, 2000). In brief, HeLa Flag-RNF4 (endogenously tagged) cells were transfected with His-Ub containing plasmid (HA-Ub for control) for 48 h and treated as indicated, followed by cell lysis with Ni-NTA pulldown lysis buffer (6 M guanidine-HCl, 100 mM $NaH_2PO_4$, 0.05% Tween20, and 100 mM Tris-HCl (pH8.0)). The His-Ub conjugates were purified by Ni-NTA beads and resuspended in Laemmli for western blotting.

## Streptavidin pulldown

HeLa WT cells were seeded in a 10 cm dish and harvested after 48 h in PBS, lysed in 300 µl RIPA buffer per sample (50 mM Tris-HCl pH 7.5, 0.1% SDS, 0.5% Sodium deoxycholate, 1 mM EDTA,

150 mM NaCl, addition of protease inhibitors freshly) and incubated for 10 min on ice. Cells were sonicated at 40% with a 1 s pulse ON and 2 s pulse OFF for 10 s to fully shear chromatin and chromatin-associated proteins. To clear the lysate, cells were centrifuged for 10 min at 16,000 × g at 4 °C, followed by performing a Lowry protein assay of the supernatant for determination of the protein concentration. About 50 µg of protein lysis was used for the Input control, and 1 mg was used for the pulldown, where biotin-CCW16 and biotin, respectively, were added in a final concentration of 10 µM and incubated for 2 h while rotating at 4 °C. Afterwards, lysates were incubated with streptavidin beads (Thermo Fisher Scientific) overnight at 4 °C. After washing once with RIPA buffer, three times with 1% (v/v) SDS, 8 M Urea in PBS, once with 1% (v/v) SDS in PBS and two times with PBS, pulled down proteins were either eluted by 2x Laemmli, including 100 µM biotin, for western blot analysis or on-bead digested for mass spectrometry evaluation (see mass spectrometry methods).

## Mass spectrometry

Whole-cell proteome (WCP) was performed in HeLa Flp in T-Rex™ RNF4 WT cells, pretreated with doxycycline (1 µg/ml) for induction of RNF4 expression for 24 h before addition of PROTACs with a final concentration of 5 µM. After 6 h of treatment, cells were washed three times with warm PBS and scraped in lysis buffer (2% SDS, 50 mM Tris/HCl pH 8.5, 10 mM TCEP, 40 mM CAA, 1 mM PMSF, 1x cOmplete™, Mini, EDTA-free Protease Inhibitor Cocktail), followed by boiling at 95 °C for 10 min, sonication for 5 min and again boiling for 5 min at 95 °C. All the following steps were performed in low-binding tubes (Eppendorf). To precipitate the proteins, the methanol/chloroform method was used. Initially, four sample volumes of ice-cold methanol, one sample volume of chloroform, and three sample volumes of water were added and mixed vigorously between every step. Afterwards, the samples were centrifuged (20,000 × g, 15 min, 4 °C) and the top layer was removed without disturbing the interphase containing the proteins. After the addition of three sample volumes of ice-cold methanol, mixing and centrifugation (20,000 × g, 10 min, 4 °C), the supernatant was removed and the pellet washed with three sample volumes of ice-cold methanol, mixed and centrifuged (20,000 × g, 10 min, 4 °C), followed by air-drying for around 10 min. The pellet was dissolved in 100 µl digestion buffer (8 M urea, 50 mM Tris/HCl, pH 8.5), mixed vigorously and incubated for 10 min at room temperature to determine the protein concentration using the BCA assay (Thermo Fisher Scientific). After dilution of the samples to a 2 M urea concentration, the assay was performed according to the manufacturer´s manual. About 50 µg of protein was digested overnight at 37 °C using LysC (10 µl/1 mg protein) and trypsin (20 µl/1 mg protein) and stopped on the next day by the addition of 1% (v/v) trifluoroacetic acid (TFA). For desalting the samples, the tC18 Sep-Pak SPE cartridges (Waters) were used. After drying the samples by speed-vac at 60 °C, peptides were resuspended in 100 mM EPPS, 10% acetonitrile (ACN), pH 8.2. The MicroBCA assay (Thermo Fisher Scientific) was performed according to the manufacturer´s manual to determine peptide concentrations. About 10 µg of peptides were labeled with the respective TMTpro™ reagent (Thermo Fisher Scientific) in 10% ACN for 1 h at room temperature. Before pooling the samples, a labeling incorporation

test (1/20th of each sample, label efficiency >98%) was performed to determine equal labeling. The addition of hydroxylamine (final concentration 0.5%) and incubating for 15 min at room temperature quenched the labeling. According to the determined labeling intensities, labeled peptides were pooled to achieve the same labeling intensity for every sample. Using the tC18 Sep-Pak SPE cartridges (Waters), samples were desalted and were removed from the remaining free TMTpro™ reagent. Peptide fractionation was performed by high-pH liquid chromatography on a micro-flow HPLC (Dionex U3000 RSLC, Thermo Fisher Scientific). Tryptic peptides were analyzed on an Orbitrap Lumos coupled to an easy nLC 1200 (Thermo Fisher Scientific) using a 35 cm long, 75 µm ID fused-silica column packed in-house with 1.9 µm C18 particles (Reprosil pur, Dr. Maisch), and kept at 50 °C using an integrated column oven (Sonation). HPLC solvents consisted of 0.1% Formic acid in water (Buffer A) and 0.1% Formic acid, 80% acetonitrile in water (Buffer B). Assuming equal amounts in each fraction, 400 ng of peptides were eluted by a non-linear gradient from 7 to 40% B over 90 min, followed by a step-wise increase to 90% B in 6 min, which was held for another 9 min. A synchronous precursor selection (SPS) multi-notch MS3 method was used in order to minimize ratio compression as previously described (McAlister et al, 2014). Full scan MS spectra (350–1400 m/z) were acquired with a resolution of 120,000 at m/z 200, maximum injection time of 100 ms and AGC target value of $4 \times 10^5$. The most intense precursors with a charge state between 2 and 6 per full scan were selected for fragmentation ("Top Speed" with a cycle time of 1.5 s) and isolated with a quadrupole isolation window of 0.7 Th. MS2 scans were performed in the Ion trap (Turbo) using a maximum injection time of 50 ms, AGC target value of $1.5 \times 10^4$ and fragmented using CID with a normalized collision energy (NCE) of 35%. SPS-MS3 scans for quantification were performed on the ten most intense MS2 fragment ions with an isolation window of 0.7 Th (MS) and 2 m/z (MS2). Ions were fragmented using HCD with an NCE of 50% and analyzed in the Orbitrap with a resolution of 50,000 at m/z 200, scan range of 100–500 m/z, AGC target value of $1.5 \times 10^5$ and a maximum injection time of 86 ms. Repeated sequencing of already acquired precursors was limited by setting a dynamic exclusion of 60 s and 7 ppm, and advanced peak determination was deactivated. All spectra were acquired in centroid mode.

Raw data were analyzed with Proteome Discoverer 2.4 (Thermo Fisher Scientific). Acquired MS3-spectra were searched against the human reference proteome (Taxonomy ID 9606) downloaded from UniProt (12-March-2020; "One Sequence Per Gene", 20531 sequences) and a collection of common contaminants (253 entries) using SequestHT, allowing a precursor mass tolerance of 7 ppm and a fragment mass tolerance of 0.5 Da after recalibration of mass errors using the Spectra RC-node applying default settings. In addition to standard dynamic (Oxidation on methionines and Met-loss at protein N-termini) and static (Carbamidomethylation on cysteines) modifications, TMT-labeling of N-termini and lysines were set as static modifications. False discovery rates were controlled using Percolator (<1% FDR on PSM level). Only PSMs with a signal-to-noise above 10 and a co-isolation below 50% were used for protein quantification after total intensity normalization. Only highly confident proteins (combined q value <0.01) were used for downstream analyses. MS data were deposited on PRIDE (Whole cell proteome of HeLa Flp in T-Rex™ RNF4 WT treated with either CCW16, 1a or 2c, PXD056951).

Biotin-CCW16 interactors were enriched by streptavidin pull-down as described above. Each pulldown was performed in triplicates and in low-binding tubes (Eppendorf), and 1 mg of protein was used per pulldown. For on-bead digestion, beads were again washed twice with 50 mM Tris/HCl, pH 8.5 and resuspended in sodium deoxycholate (SDC) buffer (2% SDC, 4 mM CAA, 1 mM TCEP in 50 mM Tris/HCl, pH 8.5) and incubated for 10 min at 95 °C for alkylation and reduction of the proteins. After cooling down to room temperature, the digestion buffer was added (Tris/HCl pH 8.5), including 0.5 µg trypsin per sample and incubated overnight at 37 °C. The digestion was stopped by adding 1% (v/v) TFA in isopropanol, and peptides were loaded onto styrene-divinyl benzene reverse phase sulfonate (SDB-RPS) polymer sorbent solid phase extraction STAGE tips for clean-up (Kulak et al, 2014). Peptides were washed once with 1% TFA (v/v) in isopropanol, followed by washing with 2% ACN and 0.2% TFA and elution in 80% ACN/1.25% ammonia. Peptides were dried by speed-vac at 60 °C and resuspended in 2% ACN and 0.1% TFA. Samples were analyzed on a Q Exactive HF coupled to an easy nLC 1200 (Thermo Fisher Scientific) using a 35 cm long, 75 µm ID fused-silica column packed in-house with 1.9 µm C18 particles (Reprosil pur, Dr. Maisch), and kept at 50 °C using an integrated column oven (Sonation). Peptides were eluted by a non-linear gradient from 4 to 28% acetonitrile over 45 min and directly sprayed into the mass spectrometer equipped with a nanoFlex ion source (Thermo Fisher Scientific). Full scan MS spectra (300–1650 m/z) were acquired in Profile mode at a resolution of 60,000 at m/z 200, a maximum injection time of 20 ms and an AGC target value of $3 \times 10^6$ charges. Up to ten most intense peptides per full scan were isolated using a 1.4 Th window and fragmented using higher energy collisional dissociation (normalized collision energy of 27). MS/MS spectra were acquired in centroid mode with a resolution of 30,000, a maximum injection time of 54 ms and an AGC target value of $1 \times 10^5$. Single-charged ions, ions with a charge state above 5 and ions with unassigned charge states were not considered for fragmentation and dynamic exclusion was set to 20 s.

Raw data were analyzed with Proteome Discoverer 2.4 (Thermo Fisher Scientific). Acquired MS2-spectra were searched against the human reference proteome (Taxonomy ID 9606) downloaded from UniProt (17-April-2022; "HoSP_OSPG_20220417.fasta", 20509 sequences) and a collection of common contaminants (253 entries) using SequestHT, allowing a precursor mass tolerance of 10 ppm and a fragment mass tolerance of 0.02 Da after recalibration of mass errors using the Spectra RC-node applying default settings. Standard dynamic (Oxidation on methionines and acetylation at protein N-termini) and static (Carbamidomethylation on cysteines) modifications were set as modifications. False discovery rates were controlled using Percolator (<1% FDR on PSM level). Only highly confident proteins (combined $q$ value <0.01) were used for downstream analyses. MS data were deposited on PRIDE (Streptavidin biotin-CCW16 pulldown to determine CCW16-modified proteins in HeLa lysate, PXD056954).

For the identification of CCW16-bound cysteine residues on RNF4, an in vitro interaction assay was performed as described below in triplicates. Before separating the sample using SDS-PAGE, 40 mM CAA was added to the SDS-sample buffer and boiled for 10 min to reduce and alkylate proteins. After separation by SDS-PAGE, samples were stained with InstantBlue and bands were cut into small pieces (1–2 mm³). Gel pieces were transferred to low-binding tubes (Eppendorf) and washed three times for 10 min each with 50 mM ammonium bicarbonate (ABC)/40% ACN. After dehydrating gel pieces for 10 min at 37 °C with 100% ACN and drying for 30 min, proteins were digested with 1 µg trypsin (in 50 mM ABC) per sample for 30 min at 4 °C. Remaining trypsin was removed, and 50 mM ABC was added and incubated on a shaker at 500 rpm overnight at 37 °C. For peptide extraction, the samples were cooled down to room temperature and 50% ACN/0.5% TFA was added and mixed for 30 min. Afterwards, the supernatant was transferred to a new low-binding tube (Eppendorf). 50% isopropanol/0.5% TFA was added to the gel pieces and mixed again for 30 min. The supernatant was pooled with the supernatant of the step before. Finally, 1% TFA in isopropanol was incubated for 10 min with the gel pieces, and the supernatant was transferred to the remaining supernatant. Peptides were loaded onto styrene-divinyl benzene reverse phasesulfonate (SDB-RPS) polymer sorbent solid phase extraction STAGE tips for clean-up (Kulak et al, 2014) and were washed once with 1% TFA (v/v) in isopropanol, followed by washing with 0.2% TFA and elution in 80% ACN/1.25% ammonia. Peptides were dried by speed-vac at 60 °C and resuspended in 2% ACN, 0.1% TFA. Samples were analyzed on a Q Exactive HF coupled to an easy nLC 1200 (Thermo Fisher Scientific) using a 35 cm long, 75 µm ID fused-silica column packed in-house with 1.9 µm C18 particles (Reprosil pur, Dr. Maisch), and kept at 50 °C using an integrated column oven (Sonation). Peptides were eluted by a non-linear gradient from 4 to 28% acetonitrile over 45 min and directly sprayed into the mass spectrometer equipped with a nanoFlex ion source (Thermo Fisher Scientific). Full scan MS spectra (300–1650 m/z) were acquired in Profile mode at a resolution of 60,000 at m/z 200, a maximum injection time of 20 ms and an AGC target value of $3 \times 10^6$ charges. Up to ten most intense peptides per full scan were isolated using a 1.4 Th window and fragmented using higher energy collisional dissociation (normalized collision energy of 27). MS/MS spectra were acquired in centroid mode with a resolution of 30,000, a maximum injection time of 54 ms and an AGC target value of $1 \times 10^5$. Single-charged ions, ions with a charge state above 5 and ions with unassigned charge states were not considered for fragmentation and dynamic exclusion was set to 20 s.

MS raw data processing was performed with MaxQuant v1.6.17.0 (Tyanova et al, 2016). Acquired spectra were searched against the sequence of RNF4, the E.coli (strain K12) reference proteome (Taxonomy ID 83333) downloaded from UniProt (17-04-2022; 4401 sequences) and a collection of common contaminants (244 entries) using the Andromeda search engine integrated in MaxQuant (Cox et al, 2011). Oxidation on methionines and acetylation at protein N-termini, as well as carbamidomethylation, CCW16 (+345.13649 Da), and 2a (+928.41933 Da) on cysteines were considered as variable modifications during the search. Identifications were filtered to obtain false discovery rates (FDR) below 1% for modification sites, peptide spectrum matches (PSM; minimum length of seven amino acids) and proteins using a target-decoy strategy (Elias and Gygi, 2007).

## Lipid peroxidation assay

About 750,000 OCI-AML2 cells were seeded 1 day before treatment in a six-well plate. On the next day, the PROTAC [2.5 µM] or the positive control cumene hydroperoxide [100 µM] was added to the

cell culture media for 2.5 and 2 h, respectively. Thirty minutes before imaging, the cells were incubated with the Image-iT® lipid sensor C11-BODIPY$^{581/591}$ [10 μM] and washed afterwards three times in ice-cold PBS to visualize lipid peroxidation by flow cytometry. The sensor can be incorporated into the membrane of living cells and shifts its fluorescence from red to green during oxidation of lipids through lipid hydroperoxides. By calculating the ratio of the reduced lipids (mean red fluorescence intensity) to the oxidized lipids (mean green fluorescence intensity), the oxidation of lipids can be determined. For imaging, the BD FACSymphony™ A5 Cell Analyzer (BD Biosciences) was used. The mean fluorescence intensities were calculated using the FlowJo™ (FlowJo™, LLC) Software for flow cytometry analysis.

## Protein expression and purification from Rosetta *E.coli*

pGEX4-T1 expression plasmids were transformed in Rosetta *E. coli* cells and GST-tagged proteins were expressed by addition of IPTG (0.5 mM) overnight at 16 °C. Cells were centrifuged, and cell pellets were resuspended in lysis buffer (PBS, 1% (v/v) Triton X-100, 1 mM DTT, 1 mM PMSF, 25 ml lysis buffer per 500 ml bacteria culture). For complete cell lysis, cells were frozen in liquid nitrogen and thawed at 37 °C three times, followed by sonication. Afterwards, cell lyses was cleared from cell debris by centrifugation (20,000 × g for 30 min at 4 °C). The supernatant was incubated with Protino® Glutathione Agarose 4B beads for 2 h at 4 °C while rotating. Afterwards, the beads were washed three times with ice-cold washing buffer (=lysis buffer) and used directly for in vitro interaction studies or stored at −20 °C with 20% glycerol.

## In vitro interaction assay

GST-based purified proteins bound to GSH-beads were incubated either with biotin-CCW16 [5 μM] (Western blot analysis) or CCW16/**2a** [10 μM] (Mass spectrometry analysis) overnight at 4 °C while rotating. Afterwards, beads were washed four times with the TPA washing buffer (30 mM TRIS pH 7.5, 100 mM NaCl, 5 mM MgCl₂, 2 mM DTT, 0.1 mg/ml BSA, 10% glycerol, and 0.01% (v/v) NP-40) and beads were either resuspended in 30 μl of 2x Laemmli for Western blot analysis or further proceed for mass spectrometry analysis (see mass spectrometry methods).

## NanoBRET™ target engagement assay

The assay was performed as described previously (Schwalm et al, 2024). In brief, full-length VHL and CRBN were obtained as plasmids cloned in frame with an N-terminal NanoLuc-fusion (kind gift from Promega). For CRBN, DDB1 was co-expressed as an additional untagged protein. The plasmids were transfected into HEK293T cells using FuGENE 4 K (Promega, E5911), and proteins were allowed to express for 20 h. Serially diluted inhibitor and NanoBRET™ VHL and CRBN Tracer (Promega, TracerDB IDs: T000018 (CRBN) and T000019 (VHL)) at the Tracer $K_D$ concentration taken from TracerDB (tracerdb.org) (Dopfer et al, 2024) were pipetted into white 384-well plates (Greiner 781207) using an Echo acoustic dispenser (Labcyte). The corresponding protein-transfected cells were added and reseeded at a density of $2 \times 10^5$ cells/mL after trypsinization and resuspending in Opti-MEM without phenol red (Life Technologies). The system was allowed to equilibrate for 3 h at 37 °C and 5% CO₂ prior to bioluminescence resonance energy transfer (BRET) measurements. To measure BRET, NanoBRET™ NanoGlo Substrate + extracellular NanoLuc Inhibitor (Promega, N2540) was added as described in the manufacturer's protocol, and filtered luminescence was measured on a PHERAstar plate reader (BMG Labtech) equipped with a luminescence filter pair (450 nm BP filter (donor) and 610 nm LP filter (acceptor)). Competitive displacement data were then graphed using GraphPad Prism 10 software using a normalized three-parameter curve fit with the following equation: $Y = 100/(1 + 10^{(X-\text{LogIC50})})$.

## Generation of cell lines stably overexpressing GPX4

Lentiviral particles were produced using a third-generation packaging system. Briefly, HEK293T cells were transfected with the X-tremeGENE HP DNA Transfection Reagent (Sigma-Aldrich, 6366546001) and a combination of plasmids: the lentiviral transfer plasmid (p442-hGPX4 mock or p442-hGPX4 WT), pMDLg/pRRE (Addgene #12251), pRSV-Rev (Addgene #12253) and pCMV-VSV-G (Addgene #8454).

Viral supernatants were harvested 48 h post-transfection, filtered (0.45 μm) and used to transduce HT-1080 cells. 48 h after transduction, the medium was replaced with fresh medium containing puromycin (1 μg/mL), and cells were maintained under selection for 1 week. All lentiviral production and transduction procedures were performed in a biosafety level 2 (BSL-2) facility.

## PROTAC and compound synthesis

All commercial chemicals and solvents were used without further purification. All reactions were performed in an inert atmosphere (Ar). Product purification was performed on a PuriFlash column chromatography system from Interchim using prepacked silica or RP C18 columns.

The synthesized compounds were characterized by $^1$H NMR, $^{13}$C NMR, and mass spectrometry (ESI). NMR spectra were measured in DMSO-$d_6$ or CD₂Cl₂ on a Bruker AV300, AV500 or DPX600 spectrometer. Chemical shifts δ are reported in parts per million (ppm). Determination of the compound purity and mass by HPLC was carried out on an Agilent 1260 Infinity II device with a 1260 DAD HS detector (G7117C; 254, 280, and 320 nm) and a LC/MSD device (G6125B). The compounds were analyzed on a Poroshell 120 EC-C18 (Agilent, 3 × 150 mm, 2.7 μm) reversed phase column using 0.1% formic acid in water (A) and 0.1% formic acid in acetonitrile (B) as a mobile phase. The following gradient was used: Method A (ESI pos. 100–1000): 0 min. 5% B—2 min. 5% B—7 min. 98% B (flow rate of 0.5 mL/min.). Method B (ESI pos. 100-1300): 0 min. 5% B—2 min. 5% B—7 min. 98% B (flow rate of 0.5 mL/min.). UV-detection was performed at 254 and 320 nm, and all compounds used for further biological characterizations showed >95% purity. HRMS measurements (method 1) were executed on a MicrOTOF qII (ESI+) using the Hystar and OTOF control software from Bruker with a Thermo Ultimate 3000 using a flow of 200 μL/min with 50% acetonitrile and 50% water with 0.1% formic acid. Lockmass correction was performed with Bruker data analysis software with carbamazepine (m/z = 237.10224 [M + H]⁺), Flunarizine (m/z = 405.21368 [M + H]⁺), and Reserpine (m/z = 609.28066 [M + H]⁺). HRMS (method2; Orbitrap) measurements were executed on an Exploris 480 Thermo

**Table 1. Flow gradient and MS/waste valve switch for HRMS method2.**

| Time | Flow | MS acquisition | Waste valve |
|------|------|----------------|-------------|
| 0 | 0.1 | off | waste |
| 0.5 | 0.1 | on | waste → MS |
| 0.6 | 0.02 | on | MS |
| 4 | 0.02 | on | MS → waste |
| 4.1 | 1 | off | waste |
| 5 | 1 | off | waste |
| 5.1 | 0.2 | off | waste |
| 5.5 | 0.2 | off | waste → MS |
| 5.8 | 0.2 | off | MS → waste |
| 6 | 0.2 | off | waste |

(Bremen, Germany) mass spectrometer equipped with a heated electrospray source (HESI) and coupled to a liquid chromatography system Vanquish VF-P10-A binary pump, VF-A10-A auto sampler, which was set to 10 °C and which was equipped with a 25 µL injection syringe and a 100 µL sample loop. Instead of a column, a 0.18 mm, 600 mm length capillary was installed within the column compartment VH-C10-A. For automated direct infusion 2.0 µL sample was injected using the flow gradient in Table 1 with 90% pure acetonitrile and 10% water with 0.1% formic acid. The flow was switched according to Table 1 from waste to the MS and back to the waste, to prevent source contamination. For monitoring, two full scan modes were selected with the following parameters. Polarity: positive; scan range: 100 to 1500 $m/z$; resolution: 480,000; AGC target: "Standard"; maximum IT: "Auto". General settings: sheath gas flow rate: 20; auxiliary gas flow rate: 5; sweep gas flow rate: 1; spray voltage: 3.5 kV; capillary temperature: 325 °C; S-lens RF level: 50; auxiliary gas heater temperature: 125 °C. For the negative mode, all values were kept instead of the spray voltage, which was set to 2.5 kV.

Synthetic procedures and analytical data of all synthesized compounds are provided in Appendix Figs. S1–S8.

### Quantification and statistical analysis

Western blots were quantified by the manufacturer´s instructions with the LI-COR Image Studio™ software. All quantified Western blot experiments were performed in three independent experiments, and a representative Western blot is shown here. All FACS experiments were analyzed by FlowJo™ (FlowJo™, LLC) and plotted in GraphPad Prism (version 8.4.2). For statistical analysis, the unpaired Student´s $t$-test was applied for Western blot quantification, viability assays, and FACS experiments.

Mass spectrometry data (whole cell proteome, streptavidin pulldown) were statistically analyzed with the Perseus software (version 1.6.15.0). For analysis of the whole cell proteome, contaminants, low confidences, and reverse entries were removed. Afterwards, all four replicates were grouped, followed by filtering for minimal four valid values in each row in at least one group. In addition, $\log_2(x)$ transformation was performed for each normalized abundance and Student´s two-sample $t$-test was applied with a Benjamini–Hochberg FDR of 0.05. For analysis of the streptavidin pulldown, contaminants, low confidences and reverse entries were removed, followed by $\log_2(x)$ transformation of the LFQ intensities. Afterwards, samples were grouped into three replicates, and every

row was filtered for minimal valid values in at least one group. In addition, imputation of missing values was performed by replacing missing values based on the normal distribution using the default settings of Perseus. For statistical analysis, Student´s two-sample $t$-test was applied with a Benjamini–Hochberg FDR of 0.05. To calculate significant hits, Microsoft Excel was used by using the following criteria: $\log_2$ (ratio) $\geq 0.58$ and $-\log_{10}$ ($p$ value) $\geq 1.3$ (proteome) and $\log_2$ (ratio) $\geq 1$ and $-\log_{10}$ ($p$ value) $\geq 1.3$ (Streptavidin pulldown).

### Bioinformatics tools

For GO Biological Process and KEGG pathway analysis of the proteome and interaction MS studies, the publicly available ShinyGO tool (version 0.80) was used, developed by a team at South Dakota State University (SDSU) (http://bioinformatics.sdstate.edu/go/). The following settings were used: FDR cutoff (0.05), Number pathways to show (10), Pathway size: Minimum (2) and Maximum (5000). The protein-coding genes of the human genome were used as background. The significantly enriched pathways were selected by FDR and ordered by fold enrichment.

### Graphics

Synopsis, Figs. 3A,C and 4A were created with BioRender.com.

## Data availability

The MS data generated in this study are deposited in the proteomics identification database (PRIDE) (https://www.ebi.ac.uk/pride). Project accession PXD056951 refers to the project name: Whole cell proteome of HeLa Flp in T-RexTM RNF4 WT treated with either CCW16, 1a or 2c. Project accession PXD056954 refers to the project name: Streptavidin biotin-CCW16 pulldown to determine CCW16-modified proteins in HeLa lysate. Raw data of Figs. 1–6 are supplied in the source data files.

The source data of this paper are collected in the following database record: biostudies:S-SCDT-10_1038-S44319-025-00593-4.

## Peer review information

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

## Acknowledgements

This work was supported by Deutsche Forschungsgemeinschaft (DFG grants MU-1764/6- Project ID-465470262 and MU-1764/7- Project ID-494535244 to S.M.; TRR387, UbiQancer -Project ID 514894665 to S.M., S.K., C.M., and J.P.F.A.; CRC1430-Project ID 424228829 to S.K.). Deutsche Krebshilfe (grant 70114823 to S.M. and grant: 70115531 to H.J.U.). BMBF Project: "PROXIDRUGS" funded by the German Federal Ministry of Education and Research to S.M., S.K., and C.M. We thank the Quantitative Proteomics Unit at IBC2 (Goethe University, Frankfurt) for supporting MS and the DFG for funding the LC-MS systems used in this study: Orbitrap Fusion LUMOS: FuGG Project ID: 403765277; QExactive HF: CRC-1177-Project ID: 259130777. S.K., M.P.S., and J.W. are grateful for support by the Structural Genomics Consortium (SGC), a registered charity (no: 1097737) that receives funds from Bayer AG, Boehringer Ingelheim, Bristol Myers Squibb, Genentech, Genome Canada through Ontario Genomics Institute, EU/EFPIA/OICR/McGill/KTH/Diamond Innovative Medicines Initiative 2 Joint Undertaking [EUbOPEN grant 875510], Janssen, Pfizer, and Takeda. This work was further supported in part by the LOEWE Center Frankfurt Cancer Institute (FCI) funded by the Hessian Ministry of Higher Education, Research and the Arts [III L5-519/03/03.001-(0015)].

## Author contributions

**Gina Gotthardt**: Data curation; Formal analysis; Validation; Investigation; Methodology; Writing—original draft; Writing—review and editing; G.G. performed all experiments except BRET and HiBiT. G.G. wrote the manuscript together with S.K. and S.M. **Janik Weckesser**: Investigation; Writing—review and editing; J.W. synthesized the CCW16-based compounds. **Georg Tascher**: Investigation; G.T. provided support for performing and analyzing mass spectrometry. **Sara Barros da Gama**: Methodology; S.BdG. and H.J.U. provided methodological support and expertise for CRISPR experiments in AML cells. **Hannah J Uckelmann**: Methodology; S.BdG. and H.J.U. provided methodological support and expertise for CRISPR experiments in AML cells. **Shibo Sun**: Methodology; S.S. and J.P.F.A. provided the GPX4-expressing cell line. **Martin P Schwalm**: Methodology; M.P.S. performed BRET and HiBit assays. **Thorsten Mosler**: Methodology; T.M. provided support for performing and analyzing mass spectrometry. **Giulio Ferrario**: Methodology; G.F. provided support for performing and analyzing mass spectrometry. **José Pedro Friedmann Angeli**: Methodology; S.S. and J.P.F.A. provided the GPX4-expressing cell line. **Christian Münch**: Methodology; C.M. provided support for performing mass spectrometry. **Stefan Knapp**: Conceptualization; Supervision; Funding acquisition; Writing—review and editing; S.M. and S.K. supervised the project. G.G., S.K. and S.M. wrote the manuscript. **Stefan Müller**: Conceptualization; Supervision; Funding acquisition; Writing—original draft; Writing—review and editing; S.M. and S.K. supervised the project. G.G., S.K. and S.M. wrote the manuscript.

Source data underlying figure panels in this paper may have individual authorship assigned. Where available, figure panel/source data authorship is listed in the following database record: biostudies:S-SCDT-10_1038-S44319-025-00593-4.

## Funding

## Disclosure and competing interests statement

The authors declare no competing interests.

# Expanded View Figures

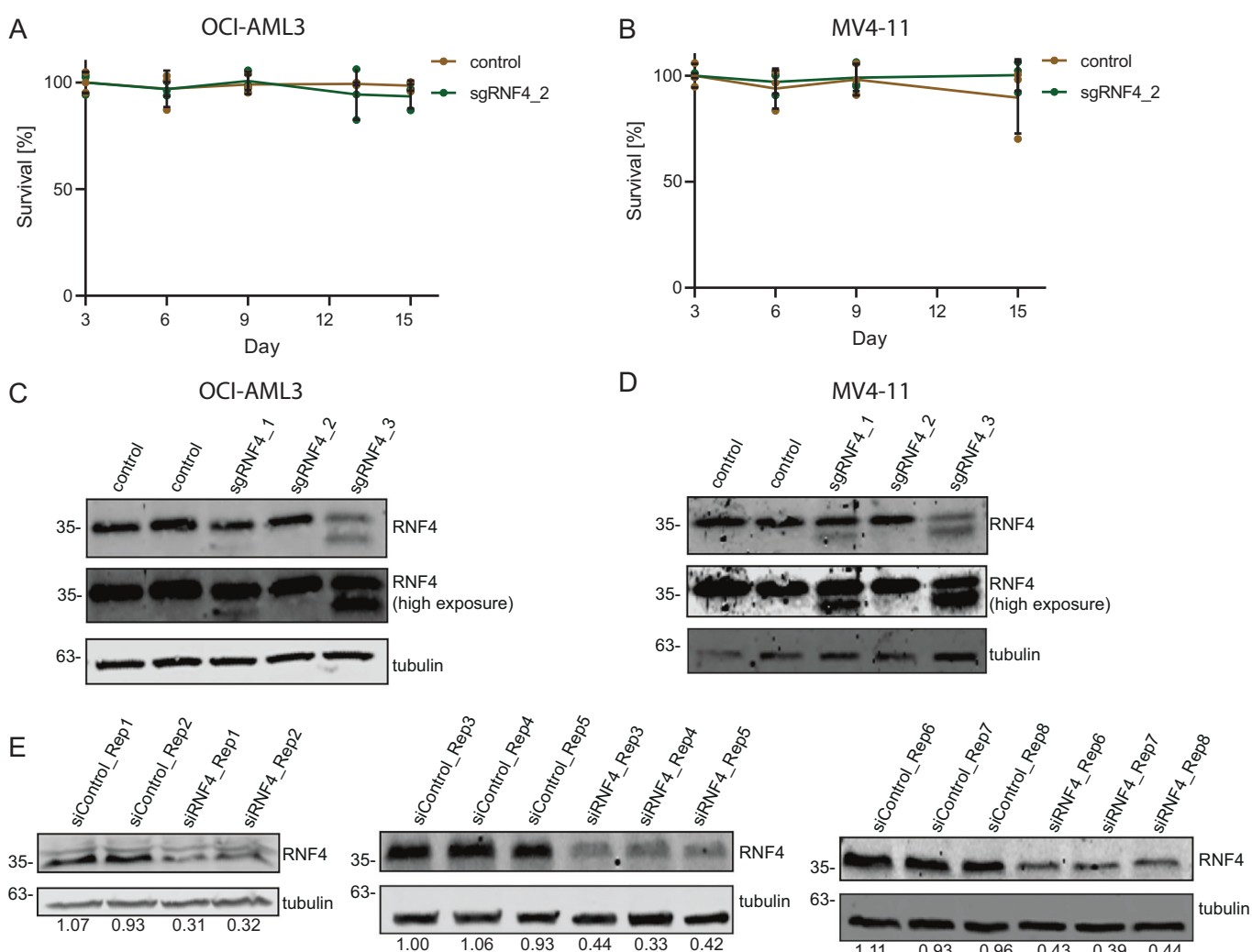

**Figure EV1. RNF4 as a vulnerability of AML cells.**

(A, B) RNF4 CRISPR dropout experiment in Cas9-expressing OCI-AML3 (A) or MV4-11 (B) cells. The transduction efficiency of PE-positive cells was around 50%. Survival rate was measured by flow cytometry and normalized to the empty vector control on day 3. Error bars show the standard deviation of the mean, $n = 3$ (biological replicates). (C, D) Confirmation of RNF4 KO in OCI-AML3 (C) and MV4-11 (D) Cas9-expressing cells after transduction with three different guideRNAs by immunoblotting. Cells were transduced and medium was exchanged after 1 day, and supplemented with 2.5 μg/ml puromycin. After 3 days (OCI-AML3) and 4 days (MV4-11), cells were harvested. Tubulin was used as loading control. (E) Validation of RNF4 KD corresponding to Fig. 1E (Rep1–Rep8) and 1F (Rep1–Rep2) by immunoblotting. Cells were harvested 3 days after the performance of KD. Tubulin was used as loading control. Quantification of the RNF4 signal, normalized to tubulin, is indicated.

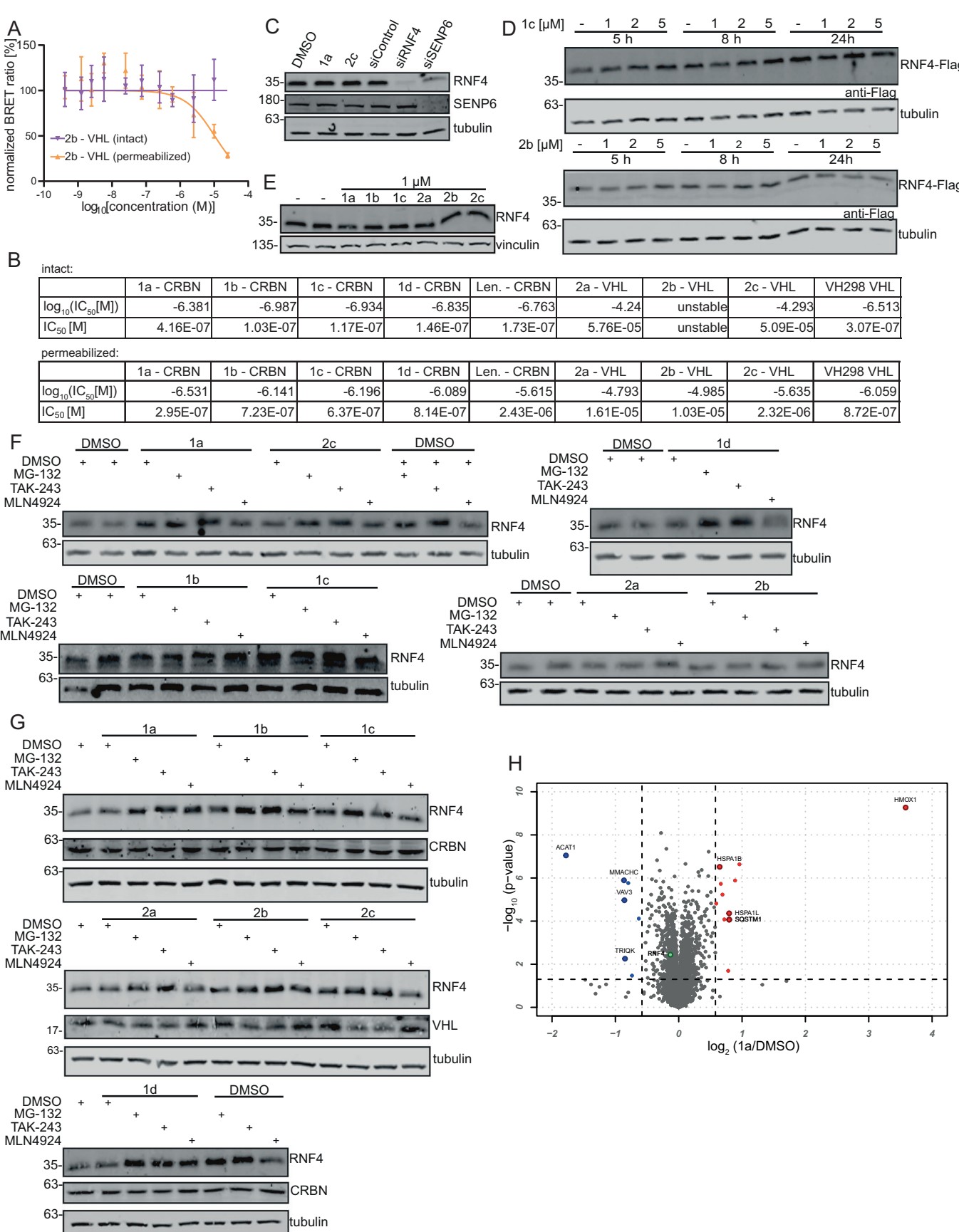

**B**

intact:

| | 1a - CRBN | 1b - CRBN | 1c - CRBN | 1d - CRBN | Len. - CRBN | 2a - VHL | 2b - VHL | 2c - VHL | VH298 VHL |
|---|---|---|---|---|---|---|---|---|---|
| log₁₀(IC₅₀[M]) | -6.381 | -6.987 | -6.934 | -6.835 | -6.763 | -4.24 | unstable | -4.293 | -6.513 |
| IC₅₀ [M] | 4.16E-07 | 1.03E-07 | 1.17E-07 | 1.46E-07 | 1.73E-07 | 5.76E-05 | unstable | 5.09E-05 | 3.07E-07 |

permeabilized:

| | 1a - CRBN | 1b - CRBN | 1c - CRBN | 1d - CRBN | Len. - CRBN | 2a - VHL | 2b - VHL | 2c - VHL | VH298 VHL |
|---|---|---|---|---|---|---|---|---|---|
| log₁₀(IC₅₀[M]) | -6.531 | -6.141 | -6.196 | -6.089 | -5.615 | -4.793 | -4.985 | -5.635 | -6.059 |
| IC₅₀ [M] | 2.95E-07 | 7.23E-07 | 6.37E-07 | 8.14E-07 | 2.43E-06 | 1.61E-05 | 1.03E-05 | 2.32E-06 | 8.72E-07 |

◀ **Figure EV2. Evaluation of RNF4 targeting PROTACs.**

(A) NanoBRET assay for PROTAC **2b** to determine cell membrane permeability in intact and permeabilized cells transiently expressing VHL with an N-terminally tagged NanoLuc. Error bars show the standard deviation of the mean, $n = 4$ (biological replicates). (B) NanoBRET assay for CRBN and VHL E3 ligand-based RNF4 PROTACs to determine cell membrane permeability. $IC_{50}$ values of Figs. 2B and EV2A are indicated. (C) HeLa WT cells were depleted of RNF4 (siRNF4) or SENP6 (siSENP6) for 72 h or were treated with CCW16-derived PROTACs [5 μM] for 6 h. Control cells were treated with DMSO. RNF4 and SENP6 levels were visualized by immunoblotting. Tubulin was used as loading control. (D) Treatment of HeLa FLAG-RNF4 (endogenously tagged) cells with different RNF4 targeting PROTACs and evaluation of RNF4 degradation level by immunoblotting. Different concentrations and time points are indicated. Control cells were treated with DMSO. Tubulin was used as loading control. (E) Evaluation of RNF4 degradation level by immunoblotting in NB-4 cells after treatment with different RNF4-targeting PROTACs. Control cells were treated with DMSO. Vinculin was used as loading control. (F) Treatment of OCI-AML2 cells with different RNF4 targeting PROTACs and evaluation of RNF4 degradation level by immunoblotting. Cells were pretreated with MG-132 [20 μM], TAK-243 [1 μM], or MLN4924 [500 nM] 30 min before PROTAC treatment [5 μM] and harvested after 6 h. Control cells were treated with DMSO. Tubulin was used as loading control. (G) Evaluation of CRBN and VHL levels in HeLa WT cells by immunoblotting after pretreating cells with MG-132 [20 μM], TAK-243 [1 μM], or MLN4924 [500 nM] 30 min before PROTAC treatment [5 μM], followed by harvesting after 6 h. DMSO was used as a control treatment, and tubulin as loading control. Same experiment is also shown in Fig. 2C. (H) Whole cell proteome analysis by mass spectrometry. HeLa cells expressing RNF4 from a doxycycline-inducible promoter were treated with **1a** [5 μM] for 6 h (same experiment as in Fig. 2E). Results of the TMT-based MS analysis are visualized in a volcano plot comparing PROTAC treatment vs. DMSO control. Hits considered as significantly upregulated are highlighted in red ($\log_2$(ratio) ≥ 0.58, $-\log_{10}$($p$ value) ≥ 1.3) and hits considered as significant downregulated are highlighted in blue ($\log_2$(ratio) ≤ −0.58, $-\log_{10}$($p$ value) ≥ 1.3). The identification of those candidates was based on two-sided Student's $t$-test analysis comparing the normalized TMT abundances of **1a** treatment with DMSO control treatment. Experiments were performed with four biological replicates.

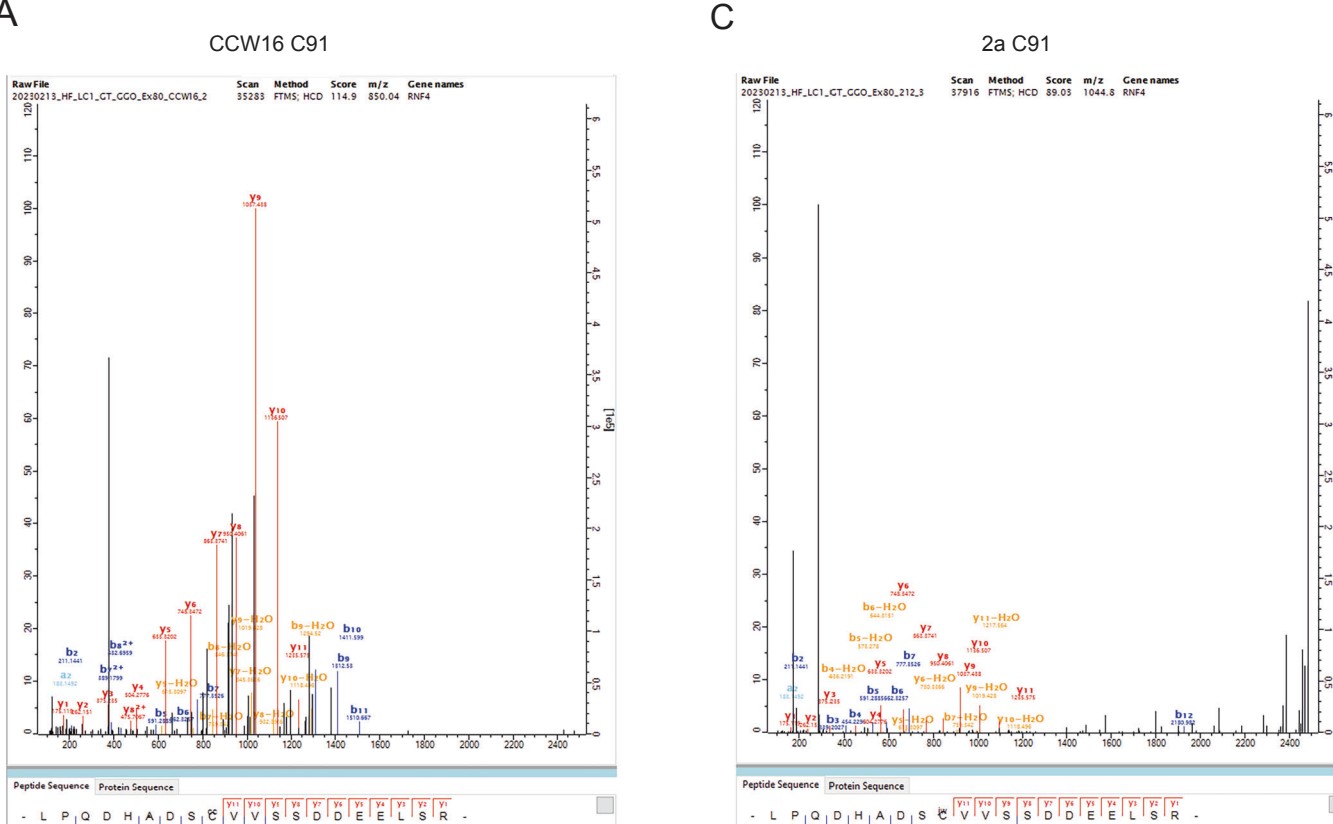

B

| Sequence | Modifications | Cysteine residues |
|----------|---------------|-------------------|
| DEGATGLR | Unmodified | - |
| DEGATGLRPSGTVSCPICMDGYSEIVQNGR | Carbamidomethyl (C) | C132, C135 |
| DEGATGLRPSGTVSCPICMDGYSEIVQNGR | 2 Carbamidomethyl (C);Oxidation (M) | C132, C135 |
| DEGATGLRPSGTVSCPICMDGYSEIVQNGR | Oxidation (M) | C132, C135 |
| DRDVYVTTHTPR | Unmodified | - |
| DVYVTTHTPR | Unmodified | - |
| GGAINSR | Unmodified | - |
| LIVSTECGHVFCSQCLR | 3 Carbamidomethyl (C) | C154, C159, C162 |
| LPQDHADSCVVSSDDEELSR | Unmodified | - |
| LPQDHADSCVVSSDDEELSR | Carbamidomethyl (C) | C91 |
| LPQDHADSCVVSSDDEELSR | 2a (C) | C91 |
| LPQDHADSCVVSSDDEELSR | CCW16 (C) | C91 |
| NANTCPTCR | 2 Carbamidomethyl (C) | C173, C176 |
| PSGTVSCPICMDGYSEIVQNGR | 2a (C) | C132, C135 |
| RLPQDHADSCVVSSDDEELSR | Carbamidomethyl (C) | C91 |
| RLPQDHADSCVVSSDDEELSR | Unmodified | C91 |
| RLPQDHADSCVVSSDDEELSR | CCW16 (C) | C91 |
| RYHPIYI | Unmodified | - |

Figure EV3.  Further evaluation of RNF4-binder CCW16 in vitro.

(A) MS spectrum of CCW16 modified GST-RNF4 on cysteine residue 91, corresponding to Fig. 3C,D. (B) Detected RNF4 peptides including the respective modifications on different cysteine residues (as indicated), corresponding to Fig. 3C,D. (C) MS spectrum of **2a** modified GST-RNF4 on cysteine residue 91, corresponding to Fig. 3C,D.

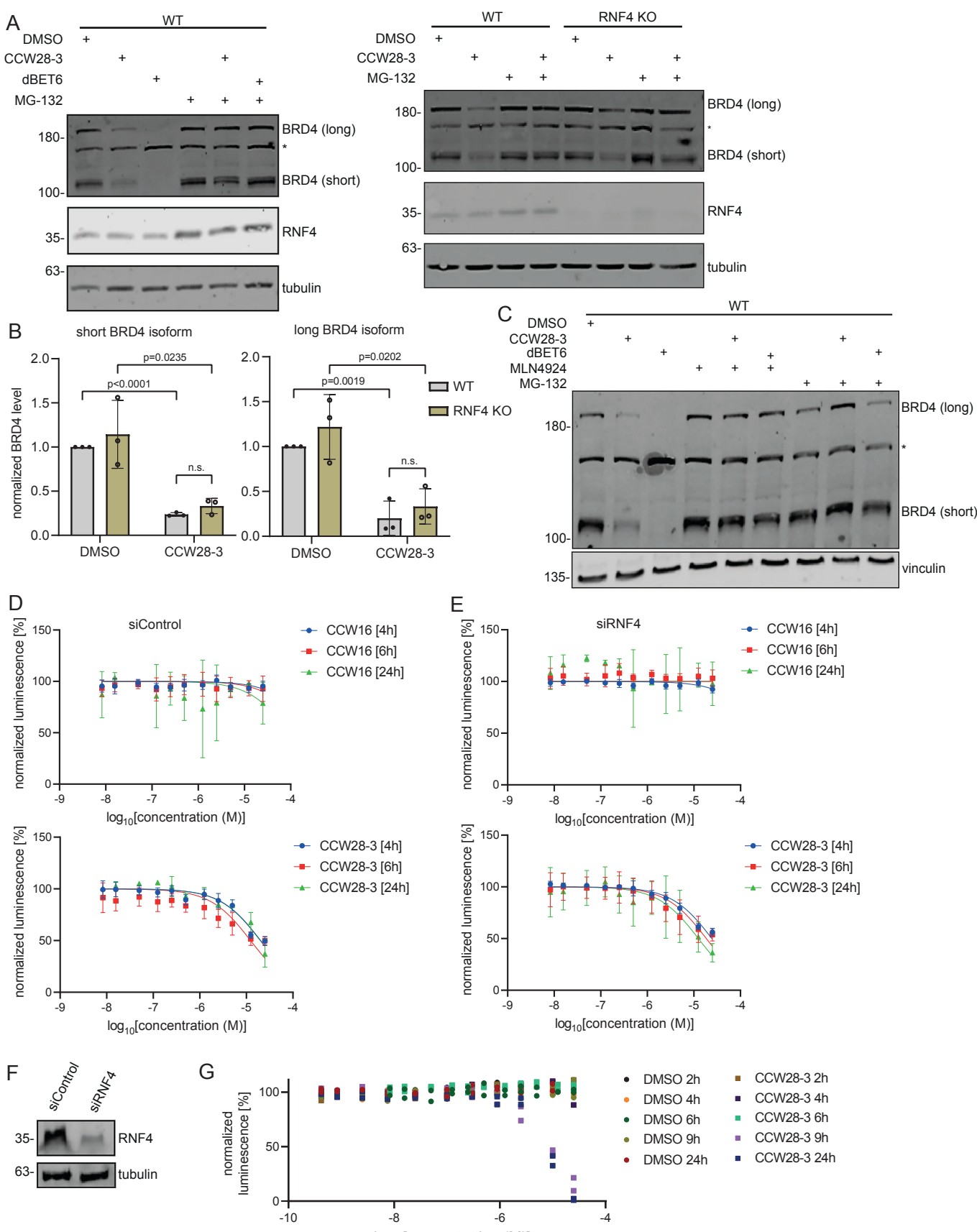

**Figure EV4.   Investigation of the RNF4-dependent BRD4 degrader JQ1-CCW16 (CCW28-3).**

(A) Pretreatment of HeLa WT cells with MG-132 [20 μM] 30 min before treatment with CCW28-3 [10 μM] or dBET6 [500 nM] and incubation for 6 h (left immunoblot). Pretreatment of HeLa WT cells or HeLa RNF4 KO cells with MG-132 [20 μM] 30 min before treatment with CCW28-3 [10 μM] or dBET6 [500 nM] and incubation for 6 h (right immunoblot). Both blots show the same experiment. Tubulin was used as loading control. *Unspecific band. (B) Quantification of BRD4 levels (short and long isoform) after CCW28-3 treatment in HeLa WT and HeLa RNF4 KO cells. Experiment was performed in three independent replicates. $P$ values of two-tailed, unpaired Student's $t$-tests are indicated; error bars show the standard deviation of the mean, $n = 3$ (biological replicates). Short isoform: WT CCW28-3 vs. DMSO: $p < 0.0001$, RNF4 KO CCW28-3 vs. DMSO: $p = 0.0235$, CCW28-3 KO vs. WT: $p = 0.1343$. Long isoform: WT CCW28-3 vs. DMSO: $p = 0.0019$, RNF4 KO CCW28-3 vs. DMSO: $p = 0.0202$, CCW28-3 KO vs. WT: $p = 0.4538$. (C) Treatment of HeLa WT cells with MG-132 [20 μM] or MLN4924 [500 nM] for 30 min followed by treatment with CCW28-3 [10 μM] or dBET6 [500 nM] for 6 h and evaluation by immunoblotting. Tubulin was used as loading control. *Unspecific band. (D, E) Measurement of BRD4 levels based on luciferase. Control KD (siControl, (D)) and RNF4 KD (siRNF4, (E)) were performed in HEK BRD4-HiBiT cells for 72 h, followed by treatment with different concentrations of CCW16 (upper panel) or CCW28-3 (lower panel) for 4, 6, and 24 h. Luciferase activity was measured by the addition of the large luciferase fragment (largeBiT) and substrate. Error bars show the standard deviation of the mean, $n = 4$ (technical replicates). (F) Confirmation of KD efficiency 3 days after performance of KD of Fig. EV4D,E by immunoblotting. (G) Evaluation of cell viability after CCW28-3 treatment in HEK BRD4-HiBiT cell lines by CellTiterGlo assay ($n = 2$, biological replicates). Concentrations and time points are indicated.

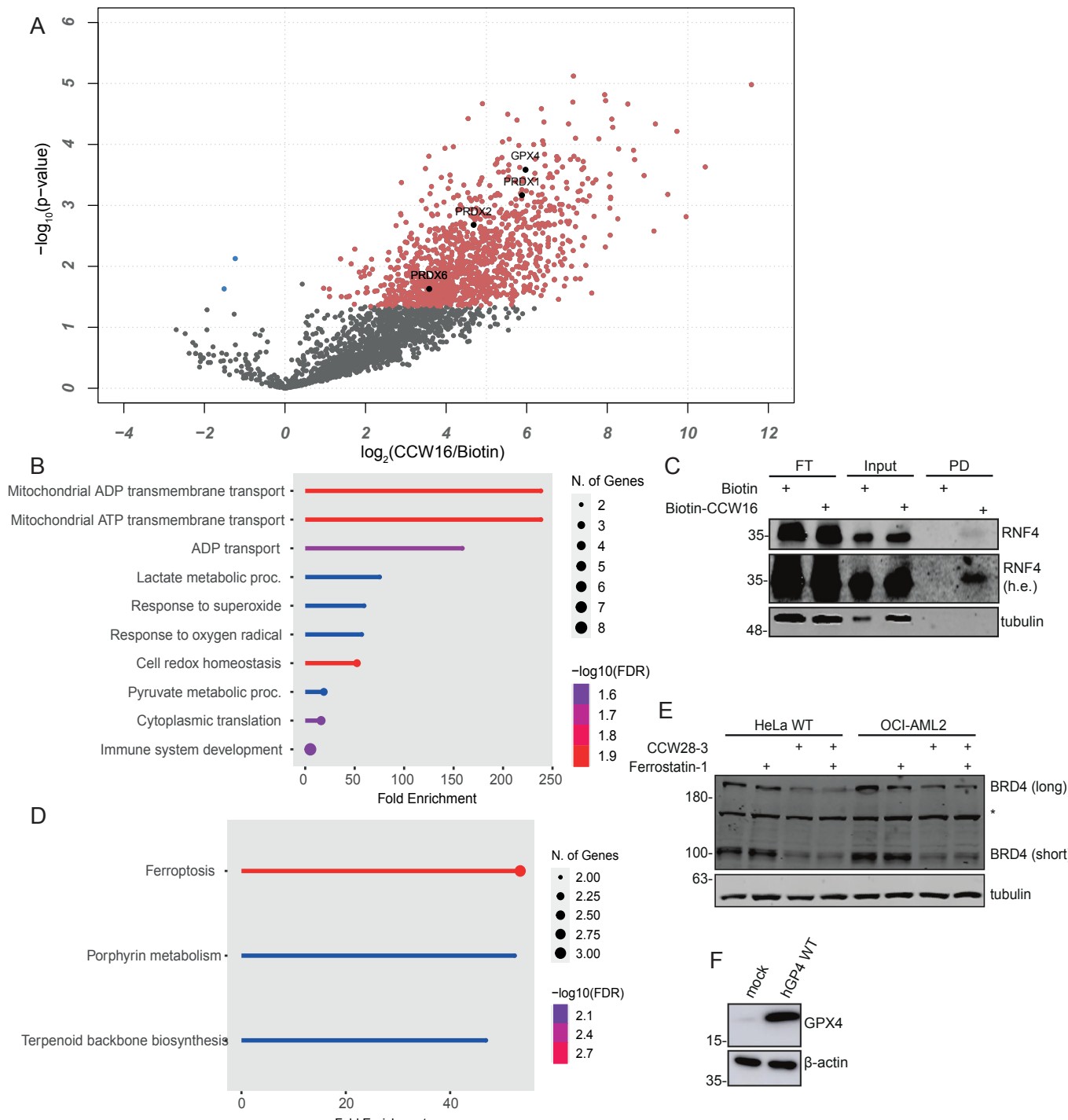

Figure EV5.    Identification of CCW16 targets in a cellular system.

(A) Volcano plot of quantitative MS analysis after biotin-CCW16 pulldown of HeLa WT cell lysates. Significantly enriched interactors are shown in red ($\log_2$(ratio) ≥ 1, $-\log_{10}$($p$ value) ≥ 1.3). Identification of candidates is based on two-sided Student's $t$-test analysis comparing LFQ intensities of biotin-CCW16 pulldown and biotin control pulldown. Experiment was performed in biological triplicates. Proteins involved in the reduction of peroxides are additionally highlighted. (B) Gene Ontology term enrichment analysis of biological processes (GOBP) of the 38 biotin-CCW16 modified proteins (from Fig. 4B) identified by MS ($\log_2$(ratio) ≥ 1, $-\log_{10}$($p$ value) ≥ 1.3). Shown here are the top ten enriched biological processes. The enrichment analysis was done using the ShinyGO tool, applying an FDR cutoff of 0.05. (C) RNF4 immunoblotting of streptavidin pulldown in HeLa WT cells. Same experiment as in Fig. 4D. Tubulin was used as loading control. FT flow through, PD pulldown, h.e. high exposure. (D) Kyoto Encyclopedia of Genes and Genomes (KEGG) pathway analysis of significantly upregulated proteins (from Fig. 2E) identified by MS ($\log_2$(ratio) ≥ 0.58, $-\log_{10}$($p$ value) ≥ 1.3). Shown here are the top three enriched biological processes. The enrichment analysis was done using the ShinyGO tool, applying an FDR cutoff of 0.05. (E) HeLa WT or OCI-AML2 cells were pretreated with ferrostatin-1 [10 μM] followed by treatment with CCW28-3 (10 μM for HeLa WT, 1 μM for OCI-AML2) for 6 h. BRD4 levels were evaluated by immunoblotting. Control cells were treated with DMSO. Tubulin was used as loading control. *Unspecific band. (F) Validation of overexpression of human GPX4 WT (hGPX4) compared to empty vector (mock) in HT-1080 cells used in Fig. 6D. β-actin was used as loading control.

