## [Peer Review File · EMBO Reports]

Cysteine-reactive covalent chloro-N-acetamide ligands induce ferroptosis mediated cell death

Gina Gotthardt, Janik Weckesser, Georg Tascher, Sara Barros da Gama, Hannah Uckelmann, Shibo Sun, Martin Schwalm, Thorsten Mosler, Giulio Ferrario, José Pedro Friedmann Angeli, Christian Münch, Stefan Knapp, and Stefan Müller

Corresponding author(s): Stefan Müller (ste.mueller@em.uni-frankfurt.de) , Stefan Knapp (knapp@pharmchem.uni-frankfurt.de)

Review Timeline:

Submission Date:	12th Dec 24
Editorial Decision:	20th Dec 24
Appeal Received:	27th Jan 25
Editorial Decision:	27th Mar 25
Revision Received:	13th Jun 25
Editorial Decision:	7th Aug 25
Revision Received:	11th Aug 25
Accepted:	28th Aug 25

Transaction Report:

Dear Stefan,

Thank you once more for the submission of your manuscript to EMBO Reports.

We acknowledge that you provide evidence that the reported covalent small molecule ligand CCW16 is not specific for RNF4. It does not bind the proposed cysteine residues 132 and 135 but shows rather unspecific binding to cysteine-containing proteins, including peroxiredoxins, thereby causing oxidative stress and ferroptosis. Moreover, the reported PROTAC CCW28-3 does degrade BRD4 but independent of RNF4.

We appreciate that you provide a comprehensive dataset that on one hand corrects the reported claims that CCW16 is an RNF4-specific ligand and that on the other hand provides evidence for its mode of action in cells. We recognize that this is important for the field.

As you might know, we consider the publication of data at EMBO Reports that refute a widely believed concept in the respective field. On occasion we also consider studies providing data that are in disagreement with a single study but these need to provide data that go beyond the refutation of an isolated claim, in this case one compound. For publication at EMBO Reports we would require further data that allow to infer a more general conclusion on the ferroptotic potential of chloro-N-acetamide containing covalent ligands - which you propose.

As the study stands, we can unfortunately not offer publication at EMBO Reports. I am very sorry to have to disappoint you on this occasion and wish you success with the rapid publication of your work.

Kind regards,

Martina

** As a service to authors, EMBO Press provides authors with the ability to transfer a manuscript that one journal cannot offer to publish to another journal, without the author having to upload the manuscript data again. To transfer your manuscript to another EMBO Press journal using this service, please click on Link Not Available

Dear Martina,

I hope you had a good start in 2025. Thanks again for the valuable feedback on our recent manuscript. We do understand your concern that we did not provide a more general view on the ferroptotic potential of chloro-N-acetamide containing covalent ligands. Considering this limitation and your feedback, we have now expanded our study and found that EN219 (a recently identified chloro-N-acetamide containing covalent ligand for RNF114 <https://www.sciencedirect.com/science/article/pii/S2451945621000052>) also exhibits a strong ferroptotic activity, most likely by targeting reactive cysteines, including cysteine residues in GPX4. Would you be willing to reconsider a revised manuscript that includes these data? In light of our new data, we would also choose a more general title and discuss the findings in a broader context.

Many thanks for your support and consideration,

Stefan

Prof. Dr. Stefan Müller

Institute of Biochemistry II • Goethe University Medical School

University Hospital Building 75 • Theodor-Stern-Kai 7 • 60590 Frankfurt, Germany

Phone ++4969 6301 83647

email: ste.mueller@em.uni-frankfurt.de

www.biochem2.de

Dear Stefan,

Thank you for the submission of your research manuscript to our journal. I apologize for the delay in handling it, but we have now received the full set of referee reports that is copied below. I have also discussed the reports and your manuscript again with the editorial team.

As you will see, the referees consider the findings potentially interesting and also value the data on unintended consequences of using chloro-N- acetamide-ligands. The referees however also raise a number of concerns that appear pertinent and need to be addressed. In particular, points 1-3 regarding the further characterization of CCW16 reactivity, specificity and broader implications for PROTAC design should be addressed.

Given the constructive comments, we would like to invite you to revise your manuscript for EMBO Reports with the understanding that the referee reports must be fully addressed and their suggestions taken on board.

Before going into the details of resubmission, I would like to make a few important points about the further handling of your manuscript that I kindly ask you to carefully consider.

We appreciate that you provide new functional insight into the role of RNF4 in protecting cells from genotoxic stress. We however also note that a major part of the manuscript reports on the off-target effects of three cysteine-reactive small molecules, CCW 16, CCW 28-3, and EN219. I agree that your findings appear relevant to the community to inform it about potential confounding mechanisms of induced protein degradation using chloro-N-acetamide ligands. That said, all of these compounds have been developed and published by the lab of Daniel Nomura and described as being specific ligands for RNF4 or RNF114. Given that your data is at odds with these earlier studies, we will be sending the revised manuscript to Dr. Nomura to give him the option to respond. A scholarly response would then be reviewed by the same referees and publication decided independently for both articles.

If you disagree with this procedure and conditions, please let me know so that we can discuss the process further. If you agree, please revise your study as outlined above. All referee concerns need be addressed in a complete point-by-point response. It is EMBO Reports policy to allow a single round of revision only and acceptance or rejection of the manuscript will therefore depend on the completeness of your responses included in the next, final version of the manuscript and on the re-review of your article in combination with the scholarly response.

We realize that it is difficult to revise to a specific deadline. In the interest of protecting the conceptual advance provided by the work, we recommend a revision within 3 months (June 27). Please discuss the revision progress ahead of this time with the editor if you require more time to complete the revisions.

I am also happy to discuss the revision further via e-mail or a video call, if you wish.

=====
IMPORTANT NOTE:

We perform an initial quality control of all revised manuscripts before re-review. Your manuscript will FAIL this control and the handling will be delayed IN CASE the following APPLIES:

- 1) A data availability section providing access to data deposited in public databases is missing. If you have not deposited any data, please add a sentence to the data availability section that explains that.
- 2) Your manuscript contains statistics and error bars based on $n=2$. Please use scatter blots in these cases. No statistics should be calculated if $n=2$.

=====

2) individual production quality figure files as .eps, .tif, .jpg (one file per figure).

Please download our Figure Preparation Guidelines (figure preparation pdf) from our Author Guidelines pages <https://www.embopress.org/page/journal/14693178/authorguide> for more info on how to prepare your figures.

4) a complete author checklist, which you can download from our author guidelines

(<<https://www.embopress.org/page/journal/14693178/authorguide>>). Please insert information in the checklist that is also reflected in the manuscript. The completed author checklist will also be part of the RPF.

5) Please note that all corresponding authors are required to supply an ORCID ID for their name upon submission of a revised manuscript (<<https://orcid.org/>>). Please find instructions on how to link your ORCID ID to your account in our manuscript tracking system in our Author guidelines

(<<https://www.embopress.org/page/journal/14693178/authorguide#authorshipguidelines>>)

6) We replaced Supplementary Information with Expanded View (EV) Figures and Tables that are collapsible/expandable online. A maximum of 5 EV Figures can be typeset. EV Figures should be cited as 'Figure EV1, Figure EV2' etc... in the text and their respective legends should be included in the main text after the legends of regular figures.

7) Please note that a Data Availability section at the end of Materials and Methods is now mandatory. In case you have no data that requires deposition in a public database, please state so instead of refereeing to the database.

See also < <https://www.embopress.org/page/journal/14693178/authorguide#dataavailability>>. Please note that the Data Availability Section is restricted to new primary data that are part of this study.

Additional information on source data and instruction on how to label the files are available

<<https://www.embopress.org/page/journal/14693178/authorguide#sourcedata>>.

10) Figure legends and data quantification:

- the name of the statistical test used to generate error bars and P values,
- the EXACT p-values,
- the number (n) of independent experiments (please specify technical or biological replicates) underlying each data point,
- the nature of the bars and error bars (s.d., s.e.m.)

- If the data are obtained from n {less than or equal to} 5, show the individual data points in addition to the SD or SEM.

- If the data are obtained from n {less than or equal to} 2, use scatter blots showing the individual data points.

Discussion of statistical methodology can be reported in the materials and methods section, but figure legends should contain a

basic description of n, P and the test applied.

11) Our journal encourages inclusion of *data citations in the reference list* to directly cite datasets that were re-used and obtained from public databases. Data citations in the article text are distinct from normal bibliographical citations and should directly link to the database records from which the data can be accessed. In the main text, data citations are formatted as follows: "Data ref: Smith et al, 2001" or "Data ref: NCBI Sequence Read Archive PRJNA342805, 2017". In the Reference list, data citations must be labeled with "[DATASET]". A data reference must provide the database name, accession number/identifiers and a resolvable link to the landing page from which the data can be accessed at the end of the reference. Further instructions are available at <<https://www.embopress.org/page/journal/14693178/authorguide#referencesformat>>.

12) All Materials and Methods need to be described in the main text using our 'Structured Methods' format. According to this format, the Methods section includes a Reagents and Tools Table (listing key reagents, experimental models, software and relevant equipment and including their sources and relevant identifiers) followed by a Methods and Protocols section describing the methods, ideally using a step-by-step protocol format. The aim is to facilitate adoption of the methodologies across labs. Please download and fill our Reagents and Tools Table template (.docx), which you can find in our author guidelines: <https://www.embopress.org/page/journal/14693178/authorguide#structuredmethods>.

13) As part of the EMBO publication's Transparent Editorial Process, EMBO Reports publishes online a Review Process File to accompany accepted manuscripts. This File will be published in conjunction with your paper and will include the referee reports, your point-by-point response and all pertinent correspondence relating to the manuscript.

Kind regards,

Martina

=====

Referee #1:

This study rigorously characterizes the anti-proliferative effects of chloroacetamide-based fragments reported to bind to RNF4. Through this work, they identify induction of ferroptosis via direct alkylation of redoxin-family proteins as the mechanism of action. Overall, this study is an important contribution that identifies potential confounding mechanisms of induced protein degradation by chloroacetamide-ligands, and assays and controls to screen for these effects.

Overall, this is an elegant study, however, some additional experiments and controls are needed.

RNF4 KO - the data in SI Fig. 1 suggests that the knockout was largely unsuccessful, and that the siRNA knockdown only produced ~ 50% reduction in RNF4 levels in AML cells (compare to HeLa cell knockout, which seems to be successful). Under these conditions, one may not expect degradation rescue. Additionally, it calls into question the RNF-4 mediated growth defects and phenotypes shown in Fig.1C and D. Can the authors address this limitation, and also confirm the genetic modifications leading to knockout by sequencing the RNF4 gene in WT and KO cells across their panel of KO cell lines.

A second missing experiment is to evaluate if the induced BRD4 degradation via an as-yet-unproposed E3-ligase is independent of ferroptosis, or a downstream event by evaluating the effect of ferroptosis inhibition on degradation.

Referee #2:

The manuscript presents an intriguing study on the unintended consequences of targeting RNF4 using covalent ligands, leading to ferroptosis instead of protein degradation. The findings are significant for both targeted protein degradation (TPD) strategies and ferroptosis research. However, in my opinion the study requires better organization and clarity, particularly in linking the different research objectives. If the authors address these issues with additional experiments and data corrections, the manuscript could be considered for publication after major revisions.

Major issues:

1. The manuscript begins with the goal of developing RNF4-targeting PROTACs but ultimately reveals an unexpected induction of ferroptosis by chloro-N-acetamide ligands. This transition needs to be more clearly articulated throughout the manuscript. While the study explores potential reasons for the failure of PROTAC-induced RNF4 degradation using mass spectrometry, proteomics, and structural modeling, it remains unclear whether RNF4 undergoes ubiquitination but is not degraded. Have the authors performed ubiquitination assays to assess this possibility? Additionally, more discussion is needed on the broader implications of this failure for covalent PROTAC development and whether such off-target effects could be leveraged for therapeutic applications.
2. CCW16 was originally identified as a covalent RNF4 binder by the Nomura group. Given its established reactivity, did the authors investigate whether its broad off-target effects were previously reported? Could a structural analysis help determine whether these effects are inherent to CCW16 or specific to the experimental context used in this study?
3. Since CCW16 was not newly developed in this study, but rather repurposed for PROTAC design, were other covalent RNF4 ligands considered? Given its broad reactivity, should alternative ligands with improved selectivity be explored for PROTAC applications?
4. The study suggests that ferroptosis is induced by CCW16-based PROTACs, but the mechanism is not fully validated. Have the authors performed GPX4 rescue experiments to confirm that ferroptosis is the cause of cell death?

Minor issues:

1. Page 5 - Figure 1E: The text states that cells were treated with the DNA-damaging compounds cytarabine, decitabine, and aphidicolin (Figure 1E). However, Figure 1E only shows decitabine and aphidicolin, with no data for cytarabine. Could the authors clarify whether cytarabine was tested but not included in the figure, or was it not part of the experiment?
2. Figure 1E - Statistical Analysis: The figure does not indicate the sample size (N) or specify the statistical analysis method used. Could the authors provide this information for clarity?
3. Synthetic Lethality Observation: The results show that RNF4 knockdown (KD) in combination with decitabine and aphidicolin induces synthetic lethality, whereas RNF4 KD alone or in combination with cytarabine does not significantly affect viability. How do the authors explain this discrepancy? If this is attributed to the transient transfection efficiency of siRNA KD, why does it still work for decitabine and aphidicolin but not for cytarabine?
4. Supplementary Figure 2A: The IC50 unit is missing from the figure. Could the authors specify the correct unit?
5. Figure 2B & Supplementary Figure 2A: The data for PROTAC 2b is missing. Could the authors clarify why it was not included?
6. Figure 2C & Supplementary Figure 2A-D: Since the authors did not observe efficient degradation of RNF4 by PROTACs, it would be beneficial to include a positive control in the Western blot experiments to validate the methodology.
7. Supplementary Figure 4 Issues: The text references Supplementary Figures 4D, 4E, and 4G, but these figures cannot be found in the supplementary material. Could the authors verify if they were omitted? Additionally, Supplementary Figure 4A and 4B do not show RNF4 degradation in RNF4 KO cells after CCW28-3 treatment, despite what is described in the text. Could the authors carefully recheck the figure and its description to ensure accuracy?
8. Missing Supplementary Table III: The file containing Supplementary Table III is not included in the provided supplementary materials. Could the authors verify the file name or check if it was inadvertently omitted?

Response to Reviewer #1 Overview of changes in main and expanded view Figures

MAIN FIGURES	
previous figure	new figure
1A	1A
1B	1B
1C	1C
1D	1D
1E	1E – modified Figure: Including more replicates to perform statistics
1F	1F
2A	2A
2B	2B
2C	2C
-	2D – new Figure: Monitoring the impact of CCW16-based PROTACs on ubiquitylation of RNF4
2D	2E – modified Figure: Volcano plot for PROTAC 1a was moved to Figure EV2H
2E	2F
3A	3A
3B	3B
3C	3C
3D	3D
3E	3E
3F	3F
4A	4A
4B	4B
4C	4C
4D	4D
5A	5A
5B	5B
5C	5C
5D	5D
5E	5E
5F	6A
5G	6B
5H	6C
	6D – new Figure: Rescuing the ferroptotic phenotype by overexpressing GPX4

EXPANDED VIEW FIGURES	
previous figure	new figure
-	1A - new Figure: Proliferation data for sgRNF4_2 in OCI-AML3
-	1B - new Figure: Proliferation data for sgRNF4_2 in MV4-11
1A	1C
1B	1D
1C	1E – modified Figure: Validation of RNF4 KD for additional replicates and quantification added
-	2A – new Figure: BRET assay of PROTAC 2b in intact and permeabilized cells
2A	2B – modified Figure: Data added for permeabilized cells
-	2C – new Figure: Control for RNF4 degradation upon RNF4 siRNA or SENP6 siRNA
2B	2D
2C	2E
2D	2F
2E	2G
	2H (from previous Fig 2D)
3A	3A
3B	3B
3C	3C
4A	now Figure EV4C
4B	now Figure EV4F
4C	now Figure EV4G
4D missing	now added as EV4A
4E missing	now added as EV4B
4F missing	now added as EV4D
4G missing	now added as EV4E
5A	5A
5B	5B
5C	5C
5D	5D
-	5E – new Figure: Monitor the impact of BRD4 degradation by CCW28-1 upon inhibition of ferroptosis
-	5F – new Figure: Control to validate expression of GPX4

The authors are grateful for the insightful comments from the reviewers. In the revised version of the manuscript, we have addressed all issues raised. A point-by-point response to each referee comment, together with the reviewer's critique, is provided below. We also provide a table giving an overview on all changes in main Figures and EV Figures.

Referee #1:

This study rigorously characterizes the anti-proliferative effects of chloroacetamide-based fragments reported to bind to RNF4. Through this work, they identify induction of ferroptosis via direct alkylation of redoxin-family proteins as the mechanism of action. Overall, this study is an important contribution that identifies potential confounding mechanisms of induced protein degradation by chloroacetamide-ligands, and assays and controls to screen for these effects.

Response: We thank the reviewer for these supportive comments.

Overall, this is an elegant study, however, some additional experiments and controls are needed.

RNF4 KO - the data in SI Fig. 1 suggests that the knockout was largely unsuccessful, and that the siRNA knockdown only produced ~ 50% reduction in RNF4 levels in AML cells (compare to HeLa cell knockout, which seems to be successful). Under these conditions, one may not expect degradation rescue. Additionally, it calls into question the RNF-4 mediated growth defects and phenotypes shown in Fig.1C and D. Can the authors address this limitation, and also confirm the genetic modifications leading to knockout by sequencing the RNF4 gene in WT and KO cells across their panel of KO cell lines.

Response: We thank the reviewer for this comment and apologize we may not have explained the CRISPR/Cas9-based KO assays shown in Figure 1C, D in sufficient detail. The data shown Figure 1C, D were not obtained with cells that exhibit a stable knock-out, because these AML cell lines were not viable upon inactivation of RNF4. We therefore had to monitor the CRISPR/Cas9-based KO at an early time point after transduction (after 3 days for OCI-AML3 and after 4 days for MV4-11), where the KO is likely not complete and where a background of non-transduced cells is still present. Nevertheless, at these time points our data in Figure EV2C, D do show truncated versions of RNF4 with the two sgRNAs (sgRNF4_1 and sgRNF4_3), which induce a growth defect, while a third sgRNA (sgRNF4_2) is inactive in both the growth assay and gene-editing. We therefore are confident that the growth defect observed over time can be attributed to RNF4 inactivation. Notably, the situation is different in HeLa cells, which tolerate depletion of RNF4. We therefore used HeLa RNF4 KO cells for the experiment shown in Figure 2F (HMOX1 induction) and the BRD4 degradation assays with CCW-28-3 (Figure EV4A). With respect to the siRNA data, we repeated the siRNA knockdown experiments and confirmed the initial data. We agree with the reviewer that knockdown was only partial. Quantification shows reduction of RNF4 levels to 30-40% (quantification for all replicates was added in Figure EV2E). As discussed in the manuscript, we conclude that under these conditions, residual RNF4 levels are sufficient to maintain normal viability under unstressed conditions, but cell viability is reduced upon exposure of cells to genotoxic stress.

We would also like to mention that reviewer #2 made us aware that in our initial version Supplementary figure 4 was missing panels 4D to 4G, although these panels were mentioned in the text section. These panels were now added as part of Figure EV4 (Panels A, B, D, E). The data show that degradation of BRD4 by CCW28-3 is independent from RNF4.

A second missing experiment is to evaluate if the induced BRD4 degradation via an as-yet-unproposed E3-ligase is independent of ferroptosis, or a downstream event by evaluating the effect of ferroptosis inhibition on degradation.

Response: We thank the reviewer for this important comment. Following the reviewer's suggestion, we now performed an experiment, where we compare the potential of CCW28-3 to degrade BRD4 in the absence or presence ferostatin in both HeLa and OCI-AML2 cells. As shown in the new Figure EV5E the addition of ferostatin does not affect BRD4 degradation indicating that degradation does not involve ferroptosis.

Response to Reviewer #2 Overview of changes in main and expanded view Figures

MAIN FIGURES	
previous figure	new figure
1A	1A
1B	1B
1C	1C
1D	1D
1E	1E – modified Figure: Including more replicates to perform statistics
1F	1F
2A	2A
2B	2B
2C	2C
-	2D – new Figure: Monitoring the impact of CCW16-based PROTACs on ubiquitylation of RNF4
2D	2E – modified Figure: Volcano plot for PROTAC 1a was moved to Figure EV2H
2E	2F
3A	3A
3B	3B
3C	3C
3D	3D
3E	3E
3F	3F
4A	4A
4B	4B
4C	4C
4D	4D
5A	5A
5B	5B
5C	5C
5D	5D
5E	5E
5F	6A
5G	6B
5H	6C
	6D – new Figure: Rescuing the ferroptotic phenotype by overexpressing GPX4

EXPANDED VIEW FIGURES	
previous figure	new figure
-	1A - new Figure: Proliferation data for sgRNF4_2 in OCI-AML3
-	1B - new Figure: Proliferation data for sgRNF4_2 in MV4-11
1A	1C
1B	1D
1C	1E – modified Figure: Validation of RNF4 KD for additional replicates and quantification added
-	2A – new Figure: BRET assay of PROTAC 2b in intact and permeabilized cells
2A	2B – modified Figure: Data added for permeabilized cells
-	2C – new Figure: Control for RNF4 degradation upon RNF4 siRNA or SENP6 siRNA
2B	2D
2C	2E
2D	2F
2E	2G
	2H (from previous Fig 2D)
3A	3A
3B	3B
3C	3C
4A	now Figure EV4C
4B	now Figure EV4F
4C	now Figure EV4G
4D missing	now added as EV4A
4E missing	now added as EV4B
4F missing	now added as EV4D
4G missing	now added as EV4E
5A	5A
5B	5B
5C	5C
5D	5D
-	5E – new Figure: Monitor the impact of BRD4 degradation by CCW28-1 upon inhibition of ferroptosis
-	5F – new Figure: Control to validate expression of GPX4

Referee #2:

The manuscript presents an intriguing study on the unintended consequences of targeting RNF4 using covalent ligands, leading to ferroptosis instead of protein degradation. The findings are significant for both targeted protein degradation (TPD) strategies and ferroptosis research. However, in my opinion the study requires better organization and clarity, particularly in linking the different research objectives. If the authors address these issues with additional experiments and data corrections, the manuscript could be considered for publication after major revisions.

Response: To make our study design easier to follow, we reorganized the abstract and provided more comprehensive descriptions of the transitions between sections as well as the rational why we included control experiments. Furthermore, we added more experimental data to the manuscript by addressing the reviewer's issues.

Major issues:

1. The manuscript begins with the goal of developing RNF4-targeting PROTACs but ultimately reveals an unexpected induction of ferroptosis by chloro-N-acetamide ligands. This transition needs to be more clearly articulated throughout the manuscript.

Response: As state above, we re-wrote parts of the manuscript. In particular, we highlight now the need for more validation assays characterizing the developed RNF4 ligands and our synthesized PROTACs. We also provide a rational for the introduced control experiments such as the rescue to the ferroptotic phenotype by overexpression of GPX4 that we included as an additional validation experiment in the revised version.

While the study explores potential reasons for the failure of PROTAC-induced RNF4 degradation using mass spectrometry, proteomics, and structural modeling, it remains unclear whether RNF4 undergoes ubiquitination but is not degraded. Have the authors performed ubiquitination assays to assess this possibility?

Response: We thank the reviewer for highlighting this. Following their suggestion, we performed an ubiquitylation assay comparing the ubiquitylation status of RNF4 in control cells and in cells treated with CCW16 and the CCW16-based PROTAC 2c. As shown in the new Figure 2D, RNF4 is ubiquitylated under basal conditions, but the level of ubiquitylation is unaltered by addition of the PROTAC 2c as well as CCW16 alone.

Additionally, more discussion is needed on the broader implications of this failure for covalent PROTAC development and whether such off-target effects could be leveraged for therapeutic applications.

Response: We have now added an additional section to the discussion describing the therapeutic potential and issues of translating this compound class into clinical or *in vivo* studies.

2. CCW16 was originally identified as a covalent RNF4 binder by the Nomura group. Given its established reactivity, did the authors investigate whether its broad off-target effects were previously reported? Could a structural analysis help determine whether these effects are inherent to CCW16 or specific to the experimental context used in this study?

Response: AlphaFold predictions as well as NMR data revealed that RNF4 is completely unstructured outside its RING domain. We therefore did not initiate any structural studies on the

RNF4 N-terminus beyond mapping the covalently modified cysteine residues to the AlphaFold model. In the discussion section, we now include what is known about the selectivity of the published RNF4 ligand. The Nomura group also identified many off-targets, in particular targets that can be linked to the cellular oxidative stress response pathway, including GPX4 itself (as seen in the supplementary MS data of their initial publication on CCW16).

3. Since CCW16 was not newly developed in this study, but rather repurposed for PROTAC design, were other covalent RNF4 ligands considered? Given its broad reactivity, should alternative ligands with improved selectivity be explored for PROTAC applications?

Response: We thank the reviewer for this comment. Currently, CCW16 is the only published and available RNF4 ligand. We demonstrated that the N-chloro-acetamide-containing EN219 ligand also strongly induces ferroptosis phenocopying the cellular response observed for CCW16. Therefore, we believe that the observed phenotype is shared within this compound class, which is important data for the drug development community, as this electrophile is frequently used in covalent ligand development. As mentioned in the discussion section, milder electrophiles might be considered for future RNF4 ligand development. However, at this stage we are concerned that the CCW16 scaffold lacks the intrinsic specificity needed for targeting RNF4.

4. The study suggests that ferroptosis is induced by CCW16-based PROTACs, but the mechanism is not fully validated. Have the authors performed GPX4 rescue experiments to confirm that ferroptosis is the cause of cell death?

Response: We thank the reviewer for this excellent suggestion. In the revised manuscript, we included now data on GPX4 overexpression, which gratifyingly completely rescued the ferroptotic phenotype protecting ectopically expressing GPX4 cells from cell death (see new Figure 6D).

Minor issues:

1. Page 5 - Figure 1E: The text states that cells were treated with the DNA-damaging compounds cytarabine, decitabine, and aphidicolin (Figure 1E). However, Figure 1E only shows decitabine and aphidicolin, with no data for cytarabine. Could the authors clarify whether cytarabine was tested but not included in the figure, or was it not part of the experiment?

Response: We initially performed the experiment with cytarabine but did not observe any synthetic effect upon RNF4 knockdown. For that reason, we finally decided to not include this data in the Figure, but by mistake did not correct the text accordingly. We assume that the different effects are due to different modes of actions of decitabine and cytarabine. In contrast to decitabine, cytarabine does not induce DNA-protein crosslinks, which are the preferred lesion of RNF4-mediated DNA repair. Further, when compared to aphidicolin, cytarabine is less potent in triggering replication stress, likely explaining the difference to aphidicolin. Altogether, we felt that the cytarabine data are of minor relevance for the current manuscript and therefore did not include it.

2. *Figure 1E - Statistical Analysis: The figure does not indicate the sample size (N) or specify the statistical analysis method used. Could the authors provide this information for clarity?*

Response: We initially performed the experiment in Figure 1E with 2 biological replicates (each consisting of 3 technical replicates). To perform a more accurate statistical analysis we repeated the experiment with 6 biological replicates (each 3 technical replicates). In the revised version, a total of 8 biological replicates were used for Figure 1E (number of replicates and statistics is detailed in the Figure legend).

3. *Synthetic Lethality Observation: The results show that RNF4 knockdown (KD) in combination with decitabine and aphidicolin induces synthetic lethality, whereas RNF4 KD alone or in combination with cytarabine does not significantly affect viability. How do the authors explain this discrepancy? If this is attributed to the transient transfection efficiency of siRNA KD, why does it still work for decitabine and aphidicolin but not for cytarabine?*

Response: see above

4. *Supplementary Figure 2A: The IC50 unit is missing from the figure. Could the authors specify the correct unit?*

Response: log(IC50) values provided have no unit. The IC50 values are shown in molar [M] (previous Figure EV2A, now Figure EV2B).

5. *Figure 2B & Supplementary Figure 2A: The data for PROTAC 2b is missing. Could the authors clarify why it was not included?*

Response: In intact cells, PROTAC 2b gave somewhat inconsistent results in the NanoBRET assay, but showed binding to VHL in permeabilized cells (see new Figures EV2A and new data added to Figure EV2B (previous Figure EV2A)). We currently have no conclusive explanation for that, but generally the VHL-based NanoBRET assay was less robust than the CRBN-based assay.

6. *Figure 2C & Supplementary Figure 2A-D: Since the authors did not observe efficient degradation of RNF4 by PROTACs, it would be beneficial to include a positive control in the Western blot experiments to validate the methodology.*

Response: We thank the reviewer for this suggestion and accordingly added further controls. By transfecting cells with anti-RNF4 siRNA we show that RNF4 levels are reduced. Further, we transfected cells with anti-SEN6 siRNA. Depletion of SEN6 was shown previously (Rojas-Fernandez et al., Molecular Cell, 2014; Wagner et al., Cell reports, 2019) to foster polySUMOylation of RNF4 leading to its autoubiquitylation and degradation. In line with this data, we do detect a reduction of RNF4 levels under these conditions, but not upon treatment of any CCW16-based PROTAC (see new Figure EV2C).

7. *Supplementary Figure 4 Issues: The text references Supplementary Figures 4D, 4E, and 4G, but these figures cannot be found in the supplementary material. Could the authors verify if they were omitted? Additionally, Supplementary Figure 4A and 4B do not show RNF4 degradation in RNF4 KO cells after CCW28-3 treatment, despite what is described in the text. Could the authors carefully recheck the figure and its description to ensure accuracy?*

Response: We apologize for this omission. As correctly stated by the reviewer Supplementary Figures 4D, 4E, 4F and 4G were missing and Supplementary Figure 4A to C also showed different figure panels as described in the text, since by mistake an earlier version of the Supplementary file was uploaded. The missing data is now included in the revised version as Figure EV4A, B, D and E. Figure EV4A and EV4B show the degradation of BRD4 after CCW28-3. The data demonstrate that degradation of BRD4 is independent of RNF4, since similar degradation effects could be detected in RNF4 KO cells. Figure EV4C shows the recovery of the BRD4 signal after inhibition of neddylation by MLN4924, which only affects cullin-dependent ligases, but not RNF4, again confirming the RNF4-independent BRD4 degradation. Figure 4D-G demonstrates again that the CCW16-based BRD4 degrader CCW28-3 indeed degrades BRD4 (as monitored here with HiBiT assays). However, degradation is unaffected by depletion of RNF4 indicating that it cannot be attributed to RNF4.

8. Missing Supplementary Table III: The file containing Supplementary Table III is not included in the provided supplementary materials. Could the authors verify the file name or check if it was inadvertently omitted?

Response: We thank the reviewer for this comment. The file was indeed omitted and is now added as dataset EV3.

Dear Stefan,

Thank you for the submission of your revised manuscript to EMBO reports. We have now received the full set of referee reports that is copied below.

As you will see, both referees consider their concerns fully addressed and recommend publication. Before I can accept the manuscript, I need you to address some minor points below:

- Please provide the specific URLs for PXD056951 and PXD056954 datasets in the data availability section of your manuscript file.
- Please place the Disclosure and competing interests statement after the Acknowledgments
- Please correct the following name discrepancies: Gina Gotthardt in the manuscript vs. Gina Gotthard in the manuscript tracking system; Sara Barros da Gama in the manuscript vs. Sara Baros da Gama in the system.
- Regarding the Author Contributions, we now use CRediT to specify the contributions of each author in the journal submission system. Therefore, please remove the Author Contributions from the manuscript file and make sure that the author contributions in our online manuscript tracking system are correct and up-to-date. The information you specified in the system will be automatically retrieved and typeset into the article. You can enter additional information in the free text box provided, if you wish.
- The funding information in the manuscript and in the manuscript tracking system is not congruent. The following also needs to be entered in the manuscript tracking system as separate funders: Orbitrap Fusion LUMOS: FuGG Project-ID: 403765277; QExactive HF: Project-ID: 259130777, SFB1177; Bayer AG, Boehringer Ingelheim, Bristol Myers Squibb, Genentech, Genome Canada through Ontario Genomics Institute, EU/EFPIA/OICR/McGill/KTH/Diamond Innovative Medicines Initiative 2 Joint Undertaking [EubOPEN grant 875510], Janssen, Pfizer and Takeda, funding by the CRC1430; LOEWE Center Frankfurt Cancer Institute (FCI) funded by the Hessian Ministry of Higher Education, Research and the Arts [III L5-519/03/03.001-(0015)]
- The Appendix needs a table of content with page numbers. Since it is a long file with a lot of data, the TOC would certainly help the reader to quickly navigate to the data of interest. The nomenclature here is Appendix Figure S1 etc. Please upload the Appendix as PDF file.
- Appendix Supplementary Methods is not a correct callout because of the "Supplementary". Please refer to the Appendix either as a whole or to specific figures.
- Reagents and Tools table: Please remove the "Instructions" text from the file, as well as the last page with examples.
- You have a second file uploaded called "Antibody list". It seems to offer additional information not given in the Reagents and Tools table. If you want to keep it as an .xls file, you could call it "Table EV1" and call it out in the Methods section. Please add a legend.
- The source data folders should not contain empty subfolders (e.g. Figure 1A). It is sufficient to only include folders for those panels for which you provide source data.
- Expanded View figure labels in the individual Figure files are not correct: they need to be changed to Figure EV1, instead of. Expanded View Figure 1, etc.
- Table 1 in the manuscript needs a legend above the table.
- BioRender should be acknowledged at the end of the Methods section in the following way:
Graphics:
(some of the... OR Figure #... OR synopsis) Graphics were created with BioRender.com.
- We perform a routine image and data integrity check on all revised manuscript and noticed the following: The RNF4 and tubulin blots shown in Figure 2C have been reused in Figure EV2G. Please confirm that the CRBN blots are indeed from the same experiment as the RNF4 and tubulin blots and please clearly indicate the reuse in both figure legends.
- Doing a spot check of the source data I noticed that in the .xls file for Figure 1E, called "Cell viability after RNF4 KD_Rep6_Rep7_Rep8" the numbers for siControl 1: Decitabine and Aphidicolin are exactly the same (rows 20 and 21 Column I-L and N-Q. Can you please double-check these numbers?
- Our production/data editors have asked you to clarify several points in the figure legends (see below). Please incorporate these changes in the manuscript and return the revised file with tracked changes with your final manuscript submission.

A) Statistical test information. Only p-values that are actually shown in the figure panel(s) should (and must) be defined in the legends, all others should be removed from (or added to) the legend. Moreover, we ask for the specification of exact p-values, unless the p-values are small ($p < 0.0001$), which can be reported as inequalities:

- Please note that the exact p values are not provided in the legends of figures 1E, 5A
- Please indicate the statistical test used for data analysis in the legends of figures 1A, B

B) Replicates and error bars:

- Please note that the box plots need to be defined in terms of minima, maxima, centre, bounds of box and whiskers, and percentile in the legend of figure 1A

- Please add the individual datapoints in addition to mean and error bars in Figure 1E.

- Please describe your findings in the Abstract in present tense.

With kind regards,

Martina

=====

Referee #1:

I thank the authors for addressing my comments in full in the revised manuscript.

Referee #2:

The authors have thoroughly addressed all of my previous concerns. I am now satisfied with the manuscript and recommend it for publication.

Points addressed in the revised version

- Please provide the specific URLs for PXD056951 and PXD056954 datasets in the data availability section of your manuscript file.

URLs for for PXD056951 and PXD056954 have been added.

- Please place the Disclosure and competing interests statement after the Acknowledgments

Done

- Please correct the following name discrepancies: Gina Gotthardt in the manuscript vs. Gina Gotthard in the manuscript tracking system; Sara Barros da Gama in the manuscript vs. Sara Baros da Gama in the system.

Done

- Regarding the Author Contributions, we now use CRediT to specify the contributions of each author in the journal submission system. Therefore, please remove the Author Contributions from the manuscript file and make sure that the author contributions in our online manuscript tracking system are correct and up-to-date. The information you specified in the system will be automatically retrieved and typeset into the article. You can enter additional information in the free text box provided, if you wish.

Done

- The funding information in the manuscript and in the manuscript tracking system is not congruent. The following also needs to be entered in the manuscript tracking system as separate funders: Orbitrap Fusion LUMOS: FuGG Project-ID: 403765277; QExactive HF: Project-ID: 259130777, SFB1177; Bayer AG, Boehringer Ingelheim, Bristol Myers Squibb, Genentech, Genome Canada through Ontario Genomics Institute, EU/EFPIA/OICR/McGill/KTH/Diamond Innovative Medicines Initiative 2 Joint Undertaking [EUbOPEN grant 875510], Janssen, Pfizer and Takeda, funding by the CRC1430; LOEWE Center Frankfurt Cancer Institute (FCI) funded by the Hessian Ministry of Higher Education, Research and the Arts [III L5-519/03/03.001-(0015)]

We have updated the funding information in the manuscript and in the manuscript tracking system.

- The Appendix needs a table of content with page numbers. Since it is a long file with a lot of data, the TOC would certainly help the reader to quickly navigate to the data of interest. The nomenclature here is Appendix Figure S1 etc. Please upload the Appendix as PDF file.

A table of content with page numbers was added.

- Appendix Supplementary Methods is not a correct callout because of the "Supplementary". Please refer to the Appendix either as a whole or to specific figures.

Has been changed.

- Reagents and Tools table: Please remove the "Instructions" text from the file, as well as the last page with examples.

Done

- You have a second file uploaded called "Antibody list". It seems to offer additional information not given in the Reagents and Tools table. If you want to keep it as an .xls file, you could call it "Table EV1" and call it out in the Methods section. Please add a legend.

Has been changed.

- The source data folders should not contain empty subfolders (e.g. Figure 1A). It is sufficient to only include folders for those panels for which you provide source data.

Empty folder was removed.

- Expanded View figure labels in the individual Figure files are not correct: they need to be changed to Figure EV1, instead of. Expanded View Figure 1, etc.

Has been changed.

- Table 1 in the manuscript needs a legend above the table.

Has been added.

- BioRender should be acknowledged at the end of the Methods section in the following way:
Graphics:
(some of the... OR Figure #... OR synopsis) Graphics were created with BioRender.com.

Has been added.

- We perform a routine image and data integrity check on all revised manuscript and noticed the following: The RNF4 and tubulin blots shown in Figure 2C have been reused in Figure EV2G. Please confirm that the CRBN blots are indeed from the same experiment as the RNF4 and tubulin blots and please clearly indicate the reuse in both figure legends.

Reuse of the blot is now indicated in the figure legend.

- Doing a spot check of the source data I noticed that in the .xls file for Figure 1E, called "Cell viability after RNF4 KD_Rep6_Rep7_Rep8" the numbers for siControl 1: Decitabine and Aphidicolin are exactly the same (rows 20 and 21 Column I-L and N-Q. Can you please double-check these numbers?

Thanks for making us aware of this error, which was indeed due to a copy/paste mistake. The numbers were correct for the Decitabine treatment, but were erroneously copied in the Aphidicolin dataset. We have corrected this and changed the bar diagram accordingly.

- Our production/data editors have asked you to clarify several points in the figure legends (see below). Please incorporate these changes in the manuscript and return the revised file with tracked changes with your final manuscript submission.

A) Statistical test information. Only p-values that are actually shown in the figure panel(s) should (and must) be defined in the legends, all others should be removed from (or added to) the legend. Moreover, we ask for the specification of exact p-values, unless the p-values are small ($p < 0.0001$), which can be reported as inequalities:

- Please note that the exact p values are not provided in the legends of figures 1E, 5A

- Please indicate the statistical test used for data analysis in the legends of figures 1A, B

Has been changed.

B) Replicates and error bars:

- Please note that the box plots need to be defined in terms of minima, maxima, centre, bounds of box and whiskers, and percentile in the legend of figure 1A

The missing information was included in the legend. Please note that the format of the violin blot has been adjusted.

- Please add the individual datapoints in addition to mean and error bars in Figure 1E.

Datapoints are added.

- Please describe your findings in the Abstract in present tense.

Done.

Prof. Stefan Müller
Goethe University Frankfurt
Medical School
Institute of Biochemistry II
Frankfurt 60590
Germany

Dear Stefan,

Thank you for implementing the final editorial requests. I am very pleased to accept your manuscript for publication in the next available issue of EMBO reports. Thank you for your contribution to our journal.

Best wishes,

Martina
